# NEURAL OPTIMAL TRANSPORT WITH GENERAL COST FUNCTIONALS

**Arip Asadulaev**[*1,3] **Alexander Korotin**[*2,1] **Vage Egiazarian**[4,5] **Petr Mokrov**[2] **Evgeny Burnaev**[2,1]

[1]Artificial Intelligence Research Institute    [2]Skolkovo Institute of Science and Technology
[3]Moscow Institute of Physics and Technology    [4]HSE University    [5]Yandex
aripasadulaev@airi.net,a.korotin@skoltech.ru

## ABSTRACT

We introduce a novel neural network-based algorithm to compute optimal transport (OT) plans for general cost functionals. In contrast to common Euclidean costs, i.e., $\ell^1$ or $\ell^2$, such functionals provide more flexibility and allow using auxiliary information, such as class labels, to construct the required transport map. Existing methods for general cost functionals are discrete and do not provide an out-of-sample estimation. We address the challenge of designing a continuous OT approach for general cost functionals in high-dimensional spaces, such as images. We construct two example functionals: one to map distributions while preserving the class-wise structure and the other one to preserve the given data pairs. Additionally, we provide the theoretical error analysis for our recovered transport plans. Our implementation is available at https://github.com/machinestein/gnot

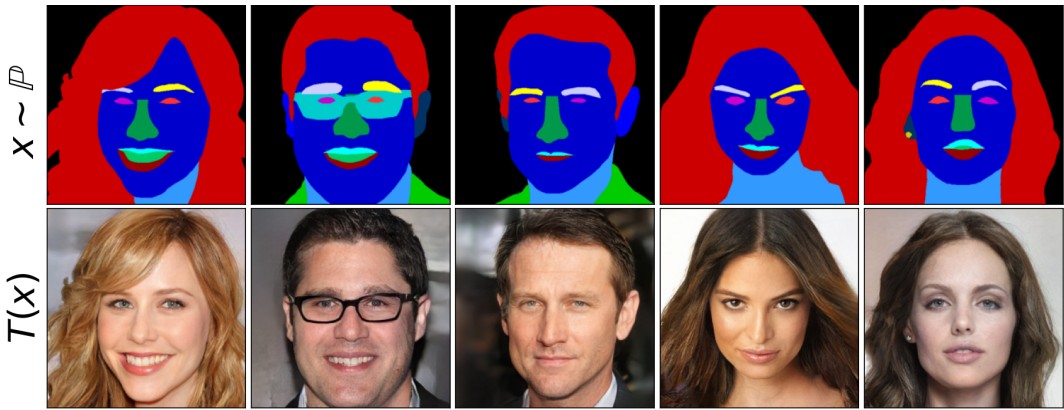

Figure 1: Results of our method with the *pair-guided cost functional* (§6.2) applied to the supervised image-to-image translation task (Celeba-MaskHQ dataset, $256 \times 256$ images).

## 1    INTRODUCTION

Optimal transport (OT) is a powerful framework to solve mass-moving problems for data distributions which finds many applications in machine learning and computer vision (Bonneel & Digne, 2023). Most existing methods to compute OT plans are designed for discrete distributions (Flamary et al., 2021; Cuturi, 2013). These methods have good flexibility and allow to control the properties of the plan (Peyré et al., 2019). However, discrete methods find an optimal matching between two given (train) sets which does not generalize to new (test) data points. This limits the use of discrete OT plan methods in scenarios where new data needs to be generated, e.g., image-to-image transfer (Zhu et al., 2017).

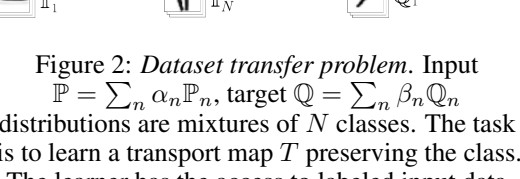

Figure 2: *Dataset transfer problem.* Input $\mathbb{P} = \sum_n \alpha_n \mathbb{P}_n$, target $\mathbb{Q} = \sum_n \beta_n \mathbb{Q}_n$ distributions are mixtures of $N$ classes. The task is to learn a transport map $T$ preserving the class. The learner has the access to labeled input data $\sim \mathbb{P}$ and only *partially labeled* target data $\sim \mathbb{Q}$.

---

*Equal contribution.

Recent works (Rout et al., 2022; Korotin et al., 2023b; 2021b; Fan et al., 2023; Daniels et al., 2021) propose **continuous** methods to compute OT plans. Thanks to employing neural networks to parameterize OT solutions, the learned transport plan can be used directly as the generative model in data synthesis (Rout et al., 2022) and unpaired learning (Korotin et al., 2023b; Rout et al., 2022; Daniels et al., 2021; Gazdieva et al., 2022).

Existing continuous OT methods mostly focus on classic cost functions such as $\ell^2$ (Korotin et al., 2021b; 2023b; Fan et al., 2023; Gazdieva et al., 2022) which estimate the closeness of input and output points. However, choosing such costs for problems where a *specific optimality of the mapping* is required may be challenging. For example, when one needs to preserve the object class during the transport (Figure 2), common $\ell^2$ cost may be suboptimal (Su et al., 2022, Appendix C), (Daniels et al., 2021, Figure 3). This limitation could be fixed by considering **general cost functionals** (Paty & Cuturi, 2020) which may take into account additional information, e.g., class labels.

Despite the large popularity of OT, the approach for continuous OT with general cost functionals (general OT) is still missing. We address this limitation. The **main contributions** of our paper are:

1. We show that the general OT problem (§2) can be reformulated as a saddle point optimization problem, which allows to implicitly recover the OT plan (§4.1) in the continuous setting. The problem can be solved with neural networks and stochastic gradient methods (Algorithm 2).

2. We provide the error analysis of solving the proposed saddle point optimization problem via the duality gaps, i.e., errors for solving inner and outer optimization problems (§4.2).

3. We construct and test examples of general cost functionals for mapping data distributions with the preservation of the class-wise (§ 5.1, Algorithm 1) and paired data structure (§5.2, Algorithm 3).

From the *theoretical* perspective, our max-min reformulation is generic and subsumes previously known reformulations for classic (Rout et al., 2022; Fan et al., 2023) and weak (Korotin et al., 2021b) OT. Furthermore, existing error analysis works exclusively with the classic OT and operate only under certain *restrictive* assumptions such as the the convexity of the dual potential. Satisfying these assumptions in practice leads to a severe performance drop (Korotin et al., 2021c, Figure 5a). In contrast, our error analysis is *free* from assumptions on the dual variable and, besides general OT, it is applicable to weak OT for which there is currently no existing error analysis.

From the *practical* perspective, we apply our method to the dataset transfer problem (Figure 2), previously not solved using continuous optimal transport. This problem arises when it is necessary to repurpose fixed or black-box models to classify previously unseen partially labelled target datasets with high accuracy by mapping the data into the dataset on which the classifier was trained (Alvarez-Melis & Fusi, 2021). Our method achieves notable improvements in accuracy over existing algorithms. Also, we show the performance of our method on the supervised image-to-image translation task.

**Notations.** For a compact Hausdorff space $\mathcal{S}$, we use $\mathcal{P}(\mathcal{S})$ to denote the set of Borel probability distributions on $\mathcal{S}$. We denote the space of continuous $\mathbb{R}$-valued functions on $\mathcal{S}$ endowed with the supremum norm by $\mathcal{C}(\mathcal{S})$. Its dual space is the space $\mathcal{M}(\mathcal{S}) \supset \mathcal{P}(\mathcal{S})$ of finite signed Borel measures over $\mathcal{S}$. Let $\mathcal{X}, \mathcal{Y}$ be compact Hausdorff spaces and $\mathbb{P} \in \mathcal{P}(\mathcal{X})$, $\mathbb{Q} \in \mathcal{P}(\mathcal{Y})$. We use $\Pi(\mathbb{P}) \subset \mathcal{P}(\mathcal{X} \times \mathcal{Y})$ to denote the subset of probability distributions on $\mathcal{X} \times \mathcal{Y}$, which projection onto the first marginal is $\mathbb{P}$. We use $\Pi(\mathbb{P}, \mathbb{Q}) \subset \Pi(\mathbb{P})$ to denote the subset of probability distributions (transport plans) on $\mathcal{X} \times \mathcal{Y}$ with marginals $\mathbb{P}, \mathbb{Q}$. For a measurable map $T : \mathcal{X} \times \mathcal{Z} \to \mathcal{Y}$, we denote the associated push-forward operator by $T_{\#}$.

## 2 BACKGROUND

In this section, we provide key concepts of the optimal transport theory. Throughout the paper, we consider compact $\mathcal{X} = \mathcal{Y} \subset \mathbb{R}^D$ and $\mathbb{P}, \mathbb{Q} \in \mathcal{P}(\mathcal{X}), \mathcal{P}(\mathcal{Y})$.

**Classic and weak OT**. For a cost function $c \in \mathcal{C}(\mathcal{X} \times \mathcal{Y})$, the *OT cost* between $\mathbb{P}, \mathbb{Q}$ is

$$\text{Cost}(\mathbb{P}, \mathbb{Q}) \stackrel{\text{def}}{=} \inf_{\pi \in \Pi(\mathbb{P}, \mathbb{Q})} \int_{\mathcal{X} \times \mathcal{Y}} c(x, y) d\pi(x, y), \tag{1}$$

see (Villani, 2008, §1). We call (1) the **classic OT**. Problem (1) admits a minimizer $\pi^* \in \Pi(\mathbb{P}, \mathbb{Q})$, which is called an *OT plan* (Santambrogio, 2015, Theorem 1.4). It may be not unique (Peyré et al., 2019, Remark 2.3). Intuitively, the cost function $c(x, y)$ measures how hard it is to move a mass piece between points $x \in \mathcal{X}$ and $y \in \mathcal{Y}$. That is, $\pi^*$ shows how to optimally distribute the mass of $\mathbb{P}$ to $\mathbb{Q}$, i.e., with minimal effort. For cost functions $c(x, y) = \|x - y\|_2$ and $c(x, y) = \frac{1}{2}\|x - y\|_2^2$,

the OT cost (1) is called the Wasserstein-1 ($\mathbb{W}_1$) and the (square of) Wasserstein-2 ($\mathbb{W}_2$) distance, respectively, see (Villani, 2008, §1) or (Santambrogio, 2015, §1, 2).

Recently, classic OT obtained the **weak OT** extension (Gozlan et al., 2017; Backhoff-Veraguas et al., 2019). Consider $C : \mathcal{X} \times \mathcal{P}(\mathcal{Y}) \to \mathbb{R}$, i.e., a weak cost function whose inputs are a point $x \in \mathcal{X}$ and a distribution of $y \in \mathcal{Y}$. The weak OT cost is

$$\text{Cost}(\mathbb{P}, \mathbb{Q}) \stackrel{\text{def}}{=} \inf_{\pi \in \Pi(\mathbb{P}, \mathbb{Q})} \int_{\mathcal{X}} C\big(x, \pi(\cdot|x)\big) d\pi(x), \qquad (2)$$

where $\pi(\cdot|x)$ denotes the conditional distribution. Weak formulation (2) is reduced to classic formulation (1) when $C(x, \mu) = \int_{\mathcal{Y}} c(x, y) d\mu(y)$. Another example of a weak cost function is the $\gamma$-weak quadratic cost $C\big(x, \mu\big) = \int_{\mathcal{Y}} \frac{1}{2} \|x - y\|_2^2 d\mu(y) - \frac{\gamma}{2} \text{Var}(\mu)$, where $\gamma \geq 0$ and $\text{Var}(\mu)$ is the variance of $\mu$, see (Korotin et al., 2023b, Eq. 5), (Alibert et al., 2019, §5.2), (Gozlan & Juillet, 2020, §5.2) for details. For this cost, we denote the optimal value of (2) by $\mathcal{W}_{2,\gamma}^2$ and call it $\gamma$-*weak Wasserstein-2*.

**Regularized and general OT**. The expression inside (1) is a linear functional. It is common to add a lower semi-continuous convex regularizer $\mathcal{R} : \mathcal{M}(\mathcal{X} \times \mathcal{Y}) \to \mathbb{R} \cup \{\infty\}$ with weight $\gamma > 0$:

$$\text{Cost}(\mathbb{P}, \mathbb{Q}) \stackrel{\text{def}}{=} \inf_{\pi \in \Pi(\mathbb{P}, \mathbb{Q})} \left\{ \int_{\mathcal{X} \times \mathcal{Y}} c(x, y) d\pi(x, y) + \gamma \mathcal{R}(\pi) \right\}. \qquad (3)$$

Regularized OT formulation (3) typically provides several advantages over original formulation (1). For example, if $\mathcal{R}(\pi)$ is strictly convex, the expression inside (3) is a strictly convex functional in $\pi$ and yields the unique OT plan $\pi^*$. Besides, regularized OT typically has better sample complexity (Genevay, 2019; Mena & Niles-Weed, 2019; Genevay et al., 2019). Common regularizers are the entropic (Cuturi, 2013), quadratic (Essid & Solomon, 2018), lasso (Courty et al., 2016), etc.

To consider a **general OT** formulation, let $\mathcal{F} : \mathcal{M}(\mathcal{X} \times \mathcal{Y}) \to \mathbb{R} \cup \{+\infty\}$ be a convex lower semi-continuous functional. Assume that there exists $\pi \in \Pi(\mathbb{P}, \mathbb{Q})$ for which $\mathcal{F}(\pi) < \infty$. Let

$$\text{Cost}(\mathbb{P}, \mathbb{Q}) \stackrel{\text{def}}{=} \inf_{\pi \in \Pi(\mathbb{P}, \mathbb{Q})} \mathcal{F}(\pi). \qquad (4)$$

This problem is a *generalization* of classic OT (1), weak OT (2), and regularized OT (3). Following (Paty & Cuturi, 2020), we call problem (4) a general OT problem. It admits a minimizer (OT plan) $\pi^*$ (Paty & Cuturi, 2020, Lemma 1). One may note that regularized OT (3) represents a similar problem: it is enough to put $c(x, y) \equiv 0$, $\gamma = 1$ and $\mathcal{R}(\pi) = \mathcal{F}(\pi)$ to obtain (4) from (3), i.e., regularized (3) and general OT (4) can be viewed as equivalent formulations.

## 3 RELATED WORK: DISCRETE AND CONTINUOUS OT SOLVERS

Solving OT problems usually implies either finding an OT plan $\pi^*$ or the OT cost. Many approaches in generative learning use *OT cost* as the loss function to update generative models, such as WGANs (Arjovsky & Bottou, 2017; Petzka et al., 2018; Liu et al., 2019), see (Korotin et al., 2022b) for a survey. These are *not related* to our work as they do not compute OT plans or maps. Existing computational OT *plan* methods can be roughly split into two groups: discrete and continuous.

**Discrete OT** considers discrete distributions $\widehat{\mathbb{P}}_N = \sum_{n=1}^{N} p_n \delta_{x_n}$ and $\widehat{\mathbb{Q}}_N = \sum_{m=1}^{M} q_m \delta_{y_m}$ and aims to find the OT plan (1), (2), (4), (3) directly between $\mathbb{P} = \widehat{\mathbb{P}}_N$ and $\mathbb{Q} = \widehat{\mathbb{Q}}_M$. In this case, the OT plan $\pi^*$ can be represented as a doubly stochastic $N \times M$ matrix. For a survey of computational methods for discrete OT, we refer to (Peyré et al., 2019). In short, one of the most popular is the Sinkhorn algorithm (Cuturi, 2013) which is designed to solve formulation (3) with the entropic regularization.

General discrete OT is extensively studied (Nash, 2000; Courty et al., 2016; Flamary et al., 2021; Ferradans et al., 2014; Rakotomamonjy et al., 2015); these methods are often employed in domain adaptation problems (Courty et al., 2016). Additionally, the available labels can be used to reconstruct the classic cost function, to capture the underlying data structure (Courty et al., 2016; Stuart & Wolfram, 2020; Liu et al., 2020; Li et al., 2019).

The major drawback of discrete OT methods is that they only perform a (stochastic) *matching* between the given empirical samples and usually *do not provide out-of-sample estimates*. This limits their application to real-world scenarios where new (test) samples frequently appear. Recent works (Hütter & Rigollet, 2021; Pooladian & Niles-Weed, 2021; Manole et al., 2021; Deb et al., 2021) consider the OT problem with the quadratic cost and develop out-of-sample estimators by wavelet/kernel-based

plugin estimators or by the barycentric projection of the discrete entropic OT plan. In spite of tractable theoretical properties, the performance of such methods in high dimensions is questionable.

**Continuous OT** usually considers $p_n = \frac{1}{N}$ and $q_n = \frac{1}{M}$ and assumes that the given discrete distributions $\widehat{\mathbb{P}}_N = \frac{1}{N}\sum_{n=1}^{N}\delta_{x_n}, \widehat{\mathbb{Q}}_M = \frac{1}{M}\sum_{m=1}^{M}\delta_{y_m}$ are the *empirical* counterparts of the underlying distributions $\mathbb{P}, \mathbb{Q}$. That is, the goal of continuous OT is to recover the OT plan between $\mathbb{P}, \mathbb{Q}$ which are accessible only by their (finite) empirical samples $\{x_1, x_2, \ldots, x_N\} \sim \mathbb{P}$ and $\{y_1, y_2, \ldots, y_M\} \sim \mathbb{Q}$. In this case, to represent the plan one has to employ parametric approximations of the OT plan $\pi^*$ or dual potentials $u^*, v^*$ which, in turn, provide *straightforward out-of-sample estimates*.

A notable development is the use of neural networks to compute OT maps for solving *weak* (2) and *classic* (1) functionals (Korotin et al., 2023b; 2022a; 2021b; Rout et al., 2022; Fan et al., 2023; Henry-Labordere, 2019). Previous OT methods were based on formulations restricted to convex potentials (Makkuva et al., 2020; Korotin et al., 2021a;c; Mokrov et al., 2021; Fan et al., 2023; Bunne et al., 2021; Alvarez-Melis et al., 2022), and used Input Convex Neural Networks (Amos et al., 2017, ICNN) to approximate them, which limited the application of OT in large-scale tasks (Korotin et al., 2021b; Fan et al., 2022; Korotin et al., 2022a). In (Genevay et al., 2016; Seguy et al., 2018; Daniels et al., 2021; Fan et al., 2022), the authors propose methods for $f$-divergence *regularized* costs (3).

While the *discrete* version of the general OT problem (4) is well studied in the literature, its continuous counterpart is not yet analyzed. **In our work**, we fill this gap by proposing the algorithm to solve the (continuous) *general* OT problem (§4), provide error bounds (§4.2). As an illustration, we construct examples of general cost functionals which can take into account the available task-specific information as labels (§5.1) or pairs (§5.2).

## 4 MAXIMIN REFORMULATION OF THE GENERAL OT

In this section, we derive a saddle point formulation for the general OT problem (4) which we later solve with neural networks. All the proofs of the statements are given in Appendix A.

### 4.1 GENERAL OT MAXIMIN REFORMULATION VIA STOCHASTIC MAPS

In this subsection, we derive the dual form for (4), which can be used to get the OT plan $\pi^*$.

Our formulation utilizes the implicit representation for plans $\Pi(\mathbb{P})$ via stochastic maps, an idea inspired by (Korotin et al., 2023b, §4.1). We introduce a latent space $\mathcal{Z} = \mathbb{R}^Z$ and an atomless distribution $\mathbb{S} \in \mathcal{P}(\mathcal{Z})$ on it, e.g., $\mathbb{S} = \mathcal{N}(0, I_Z)$. For every $\pi \in \mathcal{P}(\mathcal{X} \times \mathcal{Y})$, there exists a measurable function $T = T_\pi : \mathcal{X} \times \mathcal{Z} \to \mathcal{Y}$ which implicitly represents it. Such $T_\pi$ satisfies $T_\pi(x, \cdot)\sharp\mathbb{S} = \pi(\cdot|x)$ for all $x \in \mathcal{X}$. That is, given $x \in \mathcal{X}$ and a random latent vector $z \sim \mathbb{S}$, the function $T$ produces sample $T_\pi(x, z) \sim \pi(y|x)$. In particular, if $x \sim \mathbb{P}$, the random vector $[x, T_\pi(x, z)]$ is distributed as $\pi$. Thus, every $\pi \in \Pi(\mathbb{P})$ can be implicitly represented (non-uniquely) as a function $T_\pi : \mathcal{X} \times \mathcal{Z} \to \mathcal{Y}$. And vice-versa, every measurable function $T : \mathcal{X} \times \mathcal{Z} \to \mathcal{Y}$ is an implicit representation of the distribution $\pi_T$ which is the joint distribution of a random vector $[x, T(x, z)]$ with $x \sim \mathbb{P}, z \sim \mathbb{S}$.

Our two following theorems constitute the main theoretical idea of our approach. They are proven for separably *-increasing functionals $\mathcal{F}$ (see the Definition 1 in Appendix A). Note that one can eliminate this restiction by taking the advantage of the minimax theorems (Terkelsen, 1972).

**Theorem 1** (Maximin reformulation of the general OT). *For separably \*-increasing convex and lower semi-continuous functional* $\mathcal{F} : \mathcal{M}(\mathcal{X} \times \mathcal{Y}) \to \mathbb{R} \cup \{+\infty\}$ *it holds (we identify* $\widetilde{\mathcal{F}}(T) \stackrel{\text{def}}{=} \mathcal{F}(\pi_T)$):

$$\text{Cost}(\mathbb{P}, \mathbb{Q}) = \sup_v \inf_T \mathcal{L}(v, T) \stackrel{\text{def}}{=} \sup_v \inf_T \left\{ \widetilde{\mathcal{F}}(T) - \int_{\mathcal{X} \times \mathcal{Z}} v\big(T(x, z)\big) d\mathbb{P}(x) d\mathbb{S}(z) + \int_{\mathcal{Y}} v(y) d\mathbb{Q}(y) \right\}, \tag{5}$$

*where the* sup *is taken over potentials* $v \in \mathcal{C}(\mathcal{Y})$ *and* inf *– over measurable functions* $T : \mathcal{X} \times \mathcal{Z} \to \mathcal{Y}$.

From (5) we also see that it is enough to consider values of $\mathcal{F}$ in $\pi_T \in \Pi(\mathbb{P})$. For convention, in further derivations we always consider $\widetilde{\mathcal{F}}(T_\pi) = \mathcal{F}(\pi) = +\infty$ for $\pi \in \mathcal{M}(\mathcal{X} \times \mathcal{Y}) \setminus \Pi(\mathbb{P})$.

We say that $T^*$ is a *stochastic OT map* if it represents some OT plan $\pi^*$ solving (4), i.e., $T^*(x, \cdot)\sharp\mathbb{S} = \pi^*(\cdot|x)$ holds $\mathbb{P}$-almost surely for all $x \in \mathcal{X}$.

**Theorem 2** (Optimal saddle points provide stochastic OT maps). *Let* $v^* \in \arg\sup_v \inf_T \mathcal{L}(v, T)$ *be any optimal potential. Then for every stochastic OT map* $T^*$ *it holds:*

$$T^* \in \arg\inf_T \mathcal{L}(v^*, T). \tag{6}$$

*Furthermore, if $\mathcal{F}$ is strictly convex in $\pi$, then* (4) *permits the unique OT plan* $\pi^*$. *In this case,* $T^* \in \arg\inf_T \mathcal{L}(v^*, T) \Leftrightarrow T^*$ *is a stochastic OT map.*

From our results it follows that by solving (5) and obtaining an optimal saddle point $(v^*, T^*)$, one gets a stochastic OT map $T^*$. To ensure that all the solutions are OT maps, one may consider adding strictly convex regularizers to $\mathcal{F}$ with a small weight, e.g., conditional interaction energy, see Appendix D which is also known as the conditional kernel variance (Korotin et al., 2023a).

**Practical considerations.** Every term in (5) can be estimated with Monte Carlo by using random empirical samples from $\mathbb{P}, \mathbb{Q}$, allowing us to approach the general OT problem (4) in the continuous setting (§3). To solve the problem (5) in practice, one may use neural nets $T_\theta : \mathbb{R}^D \times \mathbb{R}^S \to \mathbb{R}^D$ and $v_\omega : \mathbb{R}^D \to \mathbb{R}$ to parametrize $T$ and $v$, respectively. To train them, one may employ stochastic gradient ascent-descent (SGAD) by using random batches from $\mathbb{P}, \mathbb{Q}, \mathbb{S}$. We summarize the optimization procedure for general cost functionals $\mathcal{F}$ in Algorithm 2 of Appendix B. In the text below (§5.1), we focus on the special cases of the class-guided functional $\mathcal{F}_G$, which is targeted to be used in the dataset transfer task (Figure 2) and pair-guided functional $\mathcal{F}_S$ for supervised image-to-image style transfer (§5.2).

**Relation to prior works.** Maximin reformulations analogous to our (5) appear in the continuous OT literature (Korotin et al., 2021c; 2023b; Rout et al., 2022; Fan et al., 2023) yet they are designed only for *classic* (1) and *weak* (2) OT. Our formulation is generic and automatically subsumes all of them. It allows using *general* cost functionals $\mathcal{F}$ which, e.g., may easily take into account side information.

### 4.2 Error Bounds for Approximate Solutions for General OT

For a pair $(\hat{v}, \hat{T})$ *approximately* solving (5), it is natural to ask how close is $\pi_{\hat{T}}$ to the OT plan $\pi^*$. Based on the duality gaps, i.e., errors for solving outer and inner optimization problems with $(\hat{v}, \hat{T})$ in (5), we give an upper bound on the difference between $\pi_{\hat{T}}$ and $\pi^*$. Our analysis holds for functionals $\mathcal{F}$ which are **strongly** convex in some metric $\rho(\cdot, \cdot)$, see Definition 2 in Appendix A. Recall that the strong convexity of $\mathcal{F}$ also implies the strict convexity, i.e., the OT plan $\pi^*$ is unique.

**Theorem 3** (Error analysis via duality gaps for stochastic maps). *Let* $\mathcal{F} : \mathcal{M}(\mathcal{X} \times \mathcal{Y}) \to \mathbb{R} \cup \{+\infty\}$ *be a convex cost functional. Let* $\rho(\cdot, \cdot)$ *be a metric on* $\Pi(\mathbb{P}) \subset \mathcal{M}(\mathcal{X} \times \mathcal{Y})$. *Assume that* $\mathcal{F}$ *is* $\beta$-*strongly convex in* $\rho$ *on* $\Pi(\mathbb{P})$. *Consider the duality gaps for an approximate solution* $(\hat{v}, \hat{T})$ *of* (5):

$$\varepsilon_1(\hat{v}, \hat{T}) \stackrel{\text{def}}{=} \mathcal{L}(\hat{v}, \hat{T}) - \inf_T \mathcal{L}(\hat{v}, T), \qquad \varepsilon_2(\hat{v}) \stackrel{\text{def}}{=} \sup_v \inf_T \mathcal{L}(v, T) - \inf_T \mathcal{L}(\hat{v}, T),$$

*which are the errors of solving the outer* $\sup_v$ *and inner* $\inf_T$ *problems in* (5), *respectively. Then for OT plan* $\pi^*$ *in* (4) *between* $\mathbb{P}$ *and* $\mathbb{Q}$ *the following inequality holds:*

$$\rho(\pi_{\hat{T}}, \pi^*) \leq \sqrt{\frac{2}{\beta}} \left( \sqrt{\varepsilon_1(\hat{v}, \hat{T})} + \sqrt{\varepsilon_2(\hat{v})} \right),$$

*i.e., the sum of the roots of duality gaps upper bounds the error of the plan* $\pi_{\hat{T}}$ *w.r.t.* $\pi^*$ *in* $\rho(\cdot, \cdot)$.

The significance of our Theorem 3 is manifested when moving from the theoretical objective (5) to its numerical counterpart. In practice, the dual potential $v$ in (5) is parameterized by NNs (a subset of continuous functions) and may not reach the optimizer $v^*$. Our duality gap analysis shows that we can still find a good approximation of the OT plan. It suffices to find a pair $(\hat{v}, \hat{T})$ that achieves *nearly* optimal objective values in the inner $\inf_T$ and outer $\sup_v$ problems of (5). In such a pair, $\pi_{\hat{T}}$ is close to the OT plan $\pi^*$. To apply our duality gap analysis, the strong convexity of $\mathcal{F}$ is required. We give an example of a strongly convex regularizer and a general recipe for using it in Appendix D. In turn, Appendix D.1 demonstrates the application of this regularization technique in practice.

**Relation to prior works.** The authors of (Fan et al., 2023), (Rout et al., 2022), (Makkuva et al., 2020) carried out error analysis via duality gaps resembling our Theorem 3. Their error analysis works *only* for classic OT (1) and requires the potential $\hat{v}$ to satisfy certain convexity properties. Our error analysis is *free* from assumptions on $\hat{v}$ and works for general OT (4) with strongly convex $\mathcal{F}$.

## 5 Learning with General Cost Functionals

In this section, we show class-guided general cost functional §5.1 for dataset transfer problem §6.1 and pair-guided cost functional §5.2 for supervised image-to-image translation §6.2.

---

**Algorithm 1:** Neural optimal transport with the class-guided cost functional $\widetilde{\mathcal{F}}_{\mathrm{G}}$.

---

**Input** : Distributions $\mathbb{P} = \sum_n \alpha_n \mathbb{P}_n$, $\mathbb{Q} = \sum_n \beta_n \mathbb{Q}_n$, $\mathbb{S}$ accessible by samples (*unlabeled*);
weights $\alpha_n$ are known and samples from each $\mathbb{P}_n, \mathbb{Q}_n$ are accessible (*labeled*);
mapping network $T_\theta : \mathbb{R}^P \times \mathbb{R}^S \to \mathbb{R}^Q$; potential network $v_\omega : \mathbb{R}^Q \to \mathbb{R}$;
number of inner iterations $K_T$;

**Output :** Learned stochastic OT map $T_\theta$ representing an OT plan between distributions $\mathbb{P}, \mathbb{Q}$;

**repeat**

  Sample (unlabeled) batches $Y \sim \mathbb{Q}$, $X \sim \mathbb{P}$ and for each $x \in X$ sample batch $Z[x] \sim \mathbb{S}$;

  $\mathcal{L}_v \leftarrow \sum\limits_{x \in X} \sum\limits_{z \in Z[x]} \frac{v_\omega(T_\theta(x,z))}{|X| \cdot |Z[x]|} - \sum\limits_{y \in Y} \frac{v_\omega(y)}{|Y|}$;

  Update $\omega$ by using $\frac{\partial \mathcal{L}_v}{\partial \omega}$;

  **for** $k_T = 1, 2, \ldots, K_T$ **do**

    Pick $n \in \{1, 2, \ldots, N\}$ at random with probabilities $(\alpha_1, \ldots, \alpha_N)$;

    Sample (labeled) batches $X_n \sim \mathbb{P}_n$, $Y_n \sim \mathbb{Q}_n$; for each $x \in X$ sample batch $Z_n[x] \sim \mathbb{S}$;

    $\mathcal{L}_T \leftarrow \widehat{\Delta \mathcal{E}^2}\big(X_n, T(X_n, Z_n), Y_n\big) - \sum\limits_{x \in X_n} \sum\limits_{z \in Z_n[x]} \frac{v_\omega(T_\theta(x,z))}{|X_n| \cdot |Z_n[x]|}$;

    Update $\theta$ by using $\frac{\partial \mathcal{L}_T}{\partial \theta}$;

**until** not converged;

---

### 5.1 CLASS-GUIDED COST FUNCTIONAL

To begin with, we theoretically formalize the problem setup. Let each input $\mathbb{P}$ and output $\mathbb{Q}$ distributions be a mixture of $N$ distributions (*classes*) $\{\mathbb{P}_n\}_{n=1}^N$ and $\{\mathbb{Q}_n\}_{n=1}^N$, respectively. That is $\mathbb{P} = \sum_{n=1}^N \alpha_n \mathbb{P}_n$ and $\mathbb{Q} = \sum_{n=1}^N \beta_n \mathbb{Q}_n$ where $\alpha_n, \beta_n \geq 0$ are the respective weights (*class prior probabilities*) satisfying $\sum_{n=1}^N \alpha_n = 1$ and $\sum_{n=1}^N \beta_n = 1$. In this general setup, we aim to find the transport plan $\pi(x, y) \in \Pi(\mathbb{P}, \mathbb{Q})$ for which the classes of $x \in \mathcal{X}$ and $y \in \mathcal{Y}$ are the same for as many pairs $(x, y) \sim \pi$ as possible. That is, its respective stochastic map $T$ should map each component $\mathbb{P}_n$ (class) of $\mathbb{P}$ to the respective component $\mathbb{Q}_n$ (class) of $\mathbb{Q}$.

The task above is related to *domain adaptation* or *transfer learning* problems. It does not always have a solution with each $\mathbb{P}_n$ exactly mapped to $\mathbb{Q}_n$ due to possible prior/posterior shift (Kouw & Loog, 2018). We aim to find a stochastic map $T$ between $\mathbb{P}$ and $\mathbb{Q}$ satisfying $T_\sharp(\mathbb{P}_n \times \mathbb{S}) \approx \mathbb{Q}_n$ for all $n = 1, \ldots, N$. To solve the above-discussed problem, we propose the following functional:

$$\mathcal{F}_{\mathrm{G}}(\pi) = \widetilde{\mathcal{F}}_{\mathrm{G}}(T_\pi) \stackrel{\text{def}}{=} \sum_{n=1}^N \alpha_n \mathcal{E}^2 \big(T_\pi \sharp(\mathbb{P}_n \times \mathbb{S}), \mathbb{Q}_n\big), \tag{7}$$

where $\mathcal{E}$ denotes the energy distance (8). For two distributions $\mathbb{Q}, \mathbb{Q}' \in \mathcal{P}(\mathcal{Y})$ with $\mathcal{Y} \subset \mathbb{R}^D$, the (square of) energy distance $\mathcal{E}$ (Rizzo & Székely, 2016) between them is:

$$\mathcal{E}^2(\mathbb{Q}, \mathbb{Q}') = \mathbb{E}\|Y_1 - Y_2\|_2 - \frac{1}{2}\mathbb{E}\|Y_1 - Y_1'\|_2 - \frac{1}{2}\mathbb{E}\|Y_2 - Y_2'\|_2, \tag{8}$$

where $Y_1 \sim \mathbb{Q}, Y_1' \sim \mathbb{Q}, Y_2 \sim \mathbb{Q}', Y_2' \sim \mathbb{Q}'$ are independent random vectors. Energy distance (8) is a particular case of the Maximum Mean Discrepancy (Sejdinovic et al., 2013). It equals zero only when $\mathbb{Q}_1 = \mathbb{Q}_2$. Hence, our functional (7) is non-negative and attains zero value when the components of $\mathbb{P}$ are correctly mapped to the respective components of $\mathbb{Q}$ (if this is possible).

**Theorem 4** (Properties of the class-guided cost functional $\mathcal{F}_{\mathrm{G}}$). *Functional $\mathcal{F}_G(\pi)$ is convex in $\pi \in \Pi(\mathbb{P})$, lower semi-continuous and $*$-separably increasing.*

In practice, each of the terms $\mathcal{E}^2\big(T_\pi \sharp(\mathbb{P}_n \times \mathbb{S}), \mathbb{Q}_n\big)$ in (7) admits estimation from samples from $\pi$.

**Proposition 1** (Estimator for $\mathcal{E}^2$). *Let $X_n \sim \mathbb{P}_n$ be a batch of $K_X$ samples from class $n$. For each $x \in X_n$ let $Z_n[x] \sim \mathbb{S}$ be a latent batch of size $K_Z$. Consider a batch $Y_n \sim \mathbb{Q}_n$ of size $K_Y$. Then*

$$\widehat{\Delta \mathcal{E}^2}\big(X_n, T(X_n, Z_n), Y_n\big) \stackrel{\text{def}}{=} \sum_{y \in Y_n} \sum_{x \in X_n} \sum_{z \in Z_n[x]} \frac{\|y - T(x,z)\|_2}{K_Y \cdot K_X \cdot K_Z} -$$
$$\sum_{x \in X_n} \sum_{z \in Z_n[x]} \sum_{x' \in X_n \setminus \{x\}} \sum_{z' \in Z_{x'}} \frac{\|T(x,z) - T(x',z')\|_2}{2 \cdot (K_X^2 - K_X) \cdot K_Z^2} \tag{9}$$

*is an estimator of $\mathcal{E}^2\big(T\sharp(\mathbb{P}_n\times\mathbb{S}),\mathbb{Q}_n\big)$ up to a constant $T$-independent shift.*

To estimate $\widetilde{\mathcal{F}}_{\mathrm{G}}(T)$, one may separately estimate terms $\mathcal{E}^2\big(T\sharp(\mathbb{P}_n\times\mathbb{S}),\mathbb{Q}_n\big)$ for each $n$ and sum them up with weights $\alpha_n$. We only estimate $n$-th term with probability $\alpha_n$ at each iteration.

We highlight the *two key details* of the estimation of (7) which are significantly different from the estimation of classic (1) and weak OT costs (2) appearing in related works (Korotin et al., 2023b; 2021b; Fan et al., 2023). First, one has to sample not just from the input distribution $\mathbb{P}$, but *separately* from each its component (class) $\mathbb{P}_n$. Moreover, one also has to be able to *separately* sample from the target distribution's $\mathbb{Q}$ components $\mathbb{Q}_n$. This is the part where the *guidance* (semi-supervision) happens. We note that to estimate costs such as classic or weak (2), no target samples from $\mathbb{Q}$ are needed at all, i.e., they can be viewed as unsupervised.

In practice, we assume that the learner is given a *labelled* empirical sample from $\mathbb{P}$ for training. In contrast, we assume that the available samples from $\mathbb{Q}$ are only *partially labelled* (with $\geq 1$ labelled data point per class). That is, we know the class label only for a limited amount of data (Figure 2). In this case, all $n$ cost terms (9) can still be stochastically estimated. These cost terms are used to learn the transport map $T_\theta$ in Algorithm 2. The remaining (unlabeled) samples will be used when training the potential $v_\omega$, as labels are not needed to update the potential in (5). We provide the detailed procedure for learning with the functional $\mathcal{F}_{\mathrm{G}}$ (7) in Algorithm 1.

## 5.2 PAIR-GUIDED COST FUNCTIONAL

In this section, we demonstrate general OT formulation with another practically-appealing specification. In particular, we define the *pair-guided* general OT cost functional. For a given paired data set $(x_1, y^*(x_1)), \ldots, (x_N, y^*(x_N))$ with samples $X_{1:N} = \{x_1, \ldots x_N\}$ and $y^*(X_{1:N}) = \{y^*(x_1), \ldots, y^*(x_N)\}$ which are assumed to follow the source $\mathbb{P}$ and target $\mathbb{Q}$ distributions, respectively, we introduce:

$$\mathcal{F}_S(\pi) \stackrel{\text{def}}{=} \int_{\mathcal{X}\times\mathcal{Y}} \ell(y, y^*(x))d\pi(x, y). \qquad (10)$$

The function $\ell : \mathcal{X} \times \mathcal{Y} \to \mathbb{R}$ is an appropriate loss measuring the difference between samples. In the majority of our experiments we choose $\ell(y, y') = \|y - y'\|_2$. In practice, to estimate (10) we assume that the learner is given a *labelled* empirical sample of $\mathbb{P}$ and $\mathbb{Q}$ for training. Using the cost (10), which can handle such information, we can train optimal mapping in a supervised manner. In (§6.2) we show how our method, together with the pair-guided functional $\mathcal{F}_S$, is directly applicable to the paired image-to-image translation problem.

## 6 EXPERIMENTAL ILLUSTRATIONS

Our Algorithm 1 is capable of learning both stochastic (*one-to-many*) $T(x, z)$ and deterministic (*one-to-one*) $T(x, z) \equiv T(x)$ transport maps. For the latter, no random noise $z$ is added to input. First, we implement stochastic and deterministic maps along with our class-guided cost function $\mathcal{F}_{\mathrm{G}}$, to address the dataset transfer problem (§6.1). Second, using the deterministic transport map $T(x)$, we apply our pair-guided functional $\mathcal{F}_S$ to solve the image-to-image supervised translation (§6.2). In the Appendix we provide experiments with toy data C.2, biological batch effect problem C.13 and various paired image-to-image datasets E. The code for the experiments can be found at



`https://github.com/machinestein/gnot`



## 6.1 CLASS-GUIDED EXPERIMENTS

**Datasets.** We use MNIST (LeCun & Cortes, 2010), FashionMNIST (Xiao et al., 2017) and MNIST-M (Ganin & Lempitsky, 2015) datasets as $\mathbb{P}, \mathbb{Q}$. Each dataset has 10 (balanced) classes and the pre-defined train-test split. In this experiment, the goal is to find a class-wise map between unrelated domains: FMNIST $\to$ MNIST and MNIST $\to$ MNIST-M. We use the default class correspondence between the datasets. For completeness, in Appendices we provide additional results with imbalanced classes (C.8), non-default correspondence (C.11), and other datasets (C).

**Baselines.** We compare our method to the pixel-level adaptation methods such as (one-to-many) AugCycleGAN (Almahairi et al., 2018; Zhu et al., 2017; Hoffman et al., 2018; Almahairi et al., 2018) and MUNIT (Huang et al., 2018; Liu et al., 2017). We use the official implementations with the hyperparameters from the respective papers. We test Neural OT (Korotin et al., 2023b; Fan et al., 2023) with Euclidean cost functions: the quadratic cost $\frac{1}{2}\|x - y\|_2^2$ ($\mathbb{W}_2$) and the $\gamma$-weak (one-to-many) quadratic cost ($\mathcal{W}_{2,\gamma}$, $\gamma = \frac{1}{10}$). For semi-supervised mapping, we considered (one-to-one)

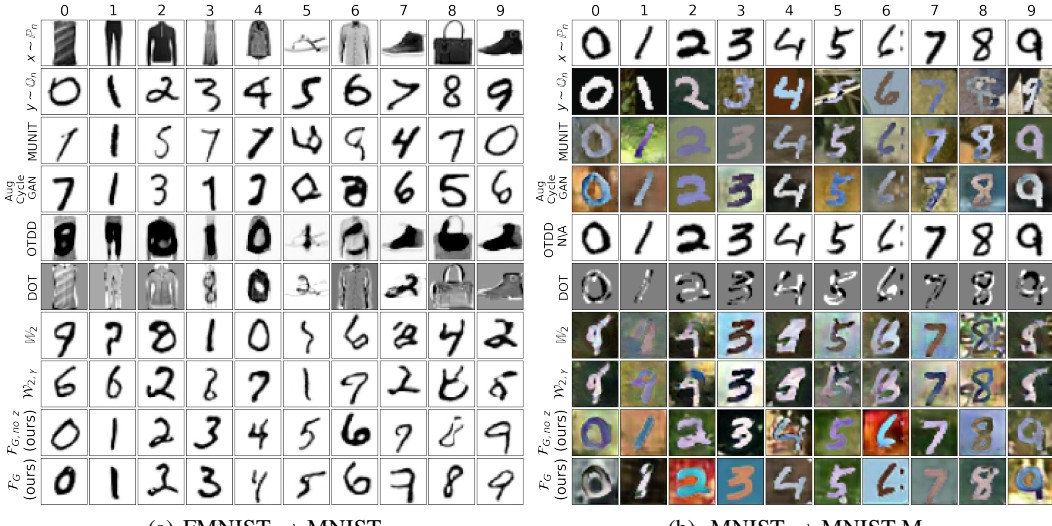

(a) FMNIST → MNIST      (b) MNIST → MNIST-M

Figure 3: The results of class-preserving mapping between two unrelated (left) and related (right) datasets. Each column shows the transfer result of a random (test) input $x \sim \mathbb{P}_n$ (first row) from a particular class ($n = 0, 1, \ldots, 9$). Each row shows the results of transfer via a particular method. For methods which learn a stochastic map $T(x, z)$, we show their output $T(x, z)$ for a random noise $z$.

| Datasets ($32 \times 32$) | Image-to-Image Translation | | Flows | Discrete OT | Neural Optimal Transport | | | |
| --- | --- | --- | --- | --- | --- | --- | --- | --- |
| | MUNIT | Aug CycleGAN | OTDD | SinkhornLpL1 | $\mathbb{W}_2$ | $\mathcal{W}_{2,\gamma}$ | $\mathcal{F}_{G}$, no $z$ [Ours] | $\mathcal{F}_{G}$ [Ours] |
| FMNIST → MNIST | 8.93 | 12.03 | 10.28 | 10.67 | 10.96 | 8.02 | 82.79 | **83.22** |
| MNIST → MNIST-M | 97.95 | **98.2** | - | 83.26 | 38.77 | 37.0 | 95.27 | 94.62 |

Table 1: Accuracy↑ of the maps learned by the translation methods in view.

| Datasets ($32 \times 32$) | MUNIT | Aug CycleGAN | OTDD | SinkhornLpL1 | $\mathbb{W}_2$ | $\mathcal{W}_{2,\gamma}$ | $\mathcal{F}_{G}$, no $z$ [Ours] | $\mathcal{F}_{G}$ [Ours] |
| --- | --- | --- | --- | --- | --- | --- | --- | --- |
| FMNIST → MNIST | 7.91 | 26.35 | > 100 | > 100 | 7.51 | 7.02 | 7.14 | **5.26** |
| MNIST → MNIST-M | 11.68 | 26.87 | - | > 100 | 19.43 | 17.48 | 18.56 | **6.67** |

Table 2: FID↓ of the samples generated by the translation methods in view.

OTDD flow (Alvarez-Melis & Fusi, 2021; 2020). This method employs gradient flows to perform the transfer preserving the class label. We also examine a General Discrete OT (DOT) which use labels. In particular, we adopted the solver from `ot.da` (Flamary et al., 2021) with its default out-of-sample estimation procedure. The solver utilizes the Sinkhorn (Cuturi, 2013) with Laplacian cost regularization (Courty et al., 2016). We show the results of *ICNN-based* OT method (Makkuva et al., 2020; Korotin et al., 2021a) in Appendix C.9.

**Metrics.** All the models are fitted on the train parts of datasets; all the provided qualitative and quantitative results are *exclusively* for test (unseen) data. To evaluate the visual quality, we compute FID (Heusel et al., 2017) of the entire mapped source test set w.r.t. the entire target test set. To estimate the accuracy of the mapping we use a pre-trained ResNet18 (He et al., 2016) classifier (with 95+ accuracy) on the target data. We consider the mapping $T$ to be correct if the predicted label for the mapped sample $T(x, z)$ matches the corresponding label of $x$.

**Results.** Qualitative results are shown in Figure 3; FID, accuracies – in Tables 2 and 1, respectively. To keep the figures simple, for all the models (one-to-one, one-to-many), we plot a single output per input. For completeness, in Appendices C.5, C.6 we show multiple outputs per each input for our method, and in Appendix C.7 we provide ablation study on $\overline{Z}$ size. Our method, general discrete OT and OTDD, use 10 labeled samples for each class in the target. Other baselines lack the capability to use label information. As seen in Figure 3 and Table 1, our approach preserves the class-wise structure accurately with just 10 labelled samples per class. The accuracy of other neural OT methods is around $10\%$, equivalent to a random guess. Both the general discrete OT and OTDD methods do not preserve the class structure in high dimensions, resulting in samples with poor FID, see table 2. Visually, the OTDD results are comparable to those in Figure 3 of (Alvarez-Melis & Fusi, 2021).

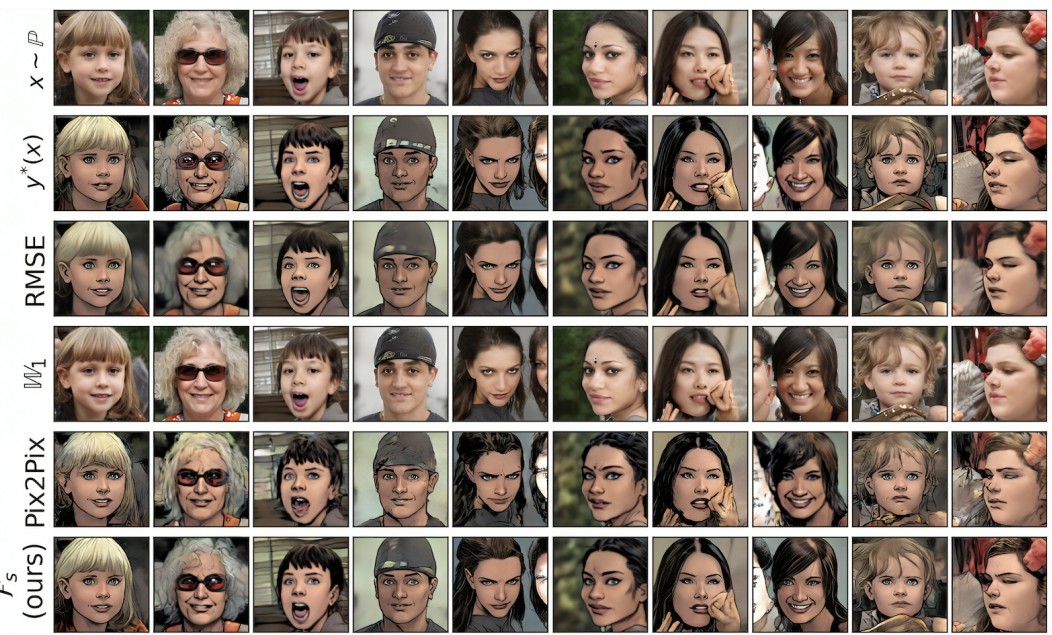

Figure 4: Results of our method with the *pair-guided cost functional* in comparison to other methods, applied to the supervised image-to-image translation task (Comic-Faces-V1, $256 \times 256$).

## 6.2 PAIR-GUIDED EXPERIMENTS

**Datasets and metrics.** We utilize three popular datasets for our evaluation: Comic-Faces-V1, Edges-to-Shoes (Isola et al., 2017), and CelebAMask-HQ (Lee et al., 2020). These datasets all contain pairs of images (handmade or formulated synthetically) and are commonly employed in benchmarking supervised image translation methods. For all experiments, we operated at the resolution of $256 \times 256$ pixels. All the results (qualitative, quantitative) are on the test sets in accordance with the default train-test splits for the datasets. As the metric, we use the test FID (as in §6.1).

**Baselines.** As the baselines, we consider the basic (unsupervised) NOT algorithm (Korotin et al., 2023b), RMSE regression, and well-celebrated supervised Pix2Pix method (Isola et al., 2017).

**Details.** In our method, we use U2Net as the transport map $T_\theta(x)$ and WGAN-QC discriminator's ResNet architecture (He et al., 2016) for potential $v_\omega$. In Comic-Faces-V1 and Edges-to-Shoes experiments, we use RMSE as the function $\ell$ in our method. In CelebAMask-HQ case, we use a VGG-based perceptual loss. Other details are given in Appendix §E.3.

**Results.** The evaluation results and comparisons with the baselines for the Comic-Faces-V1 and Edges-to-Shoes datasets are presented in Figure 4 and Figure 21, respectively. Additionally, the computed FID scores for these datasets are detailed in Table 7 in the appendix. For CelebAMask-HQ dataset, we achieve the FID score of 21.1. The examples of generated images are provided in Figures 1 and 22. Further qualitative experimentation is conducted on the Comic-Faces-V1 dataset at a higher resolution of $512 \times 512$, see Figure 23. Overall, the obtained results show that our method achieves competitive quality and can be further applied to high quality generation and editing tasks.

## 7 DISCUSSION

Our method is a generic tool to learn transport maps between data distributions with a task-specific cost functional $\mathcal{F}$. In general, the *potential impact* of our work on society depends on the scope of its application in digital content creation. As a *limitation*, we can consider the fact that to apply our method, one has to provide an estimator $\widehat{\mathcal{F}}(T)$ for the functional $\mathcal{F}$ which may be non-trivial. Besides, the construction of a cost functional $\mathcal{F}$ for a particular downstream task may be not straightforward. This should be taken into account when using the method in practice. Constructing task-specific functionals $\mathcal{F}$ and estimators $\widehat{\mathcal{F}}$ is a promising future research avenue.

## 8 ACKNOWLEDGEMENT

The work was supported by the Analytical center under the RF Government (subsidy agreement 000000D730321P5Q0002, Grant No. 70-2021-00145 02.11.2021).

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

## A PROOFS

### A.1 PROOFS OF RESULTS OF §4

To begin with, we recall some basics which play an important role in the further derivations.

**Conjugate functional.** Let $\mathcal{F} : \mathcal{M}(\mathcal{X} \times \mathcal{Y}) \to \mathbb{R} \cup \{\infty\}$ be a functional. The convex conjugate functional of $\mathcal{F}$ is $\mathcal{F}^* : C(\mathcal{X} \times \mathcal{Y}) \to \mathbb{R} \cup \{\infty\}$:

$$\mathcal{F}^*(h) \stackrel{\text{def}}{=} \sup_{\pi \in \mathcal{M}(\mathcal{X} \times \mathcal{Y})} \left[ \int_{\mathcal{X} \times \mathcal{Y}} h(s) d\pi(s) - \mathcal{F}(\pi) \right].$$

Let $u, v \in \mathcal{C}(\mathcal{X}), \mathcal{C}(\mathcal{Y})$. We write $u \oplus v \in \mathcal{C}(\mathcal{X} \times \mathcal{Y})$ to denote the function $u \oplus v : (x, y) \mapsto u(x) + v(y)$. The next definition is borrowed from (Paty & Cuturi, 2020, Definition 2).

**Definition 1** (Separably *-increasing functional). *For $\mathcal{F} : \mathcal{M}(\mathcal{X} \times \mathcal{Y}) \to \mathbb{R}$ we say that it is separably *-increasing if for all functions $u, v \in \mathcal{C}(\mathcal{X}), \mathcal{C}(\mathcal{Y})$ and any function $c \in \mathcal{C}(\mathcal{X} \times \mathcal{Y})$ from $u \oplus v \leq c$ (point-wise) it follows $\mathcal{F}^*(u \oplus v) \leq \mathcal{F}^*(c)$.*

*Proof of Theorem 1.* Since $\mathcal{F}$ is convex, lsc and separably *-increasing functional, general OT problem (4) permits the following dual representation (Paty & Cuturi, 2020, Theorem 2):

$$\text{Cost}(\mathbb{P}, \mathbb{Q}) = \sup_{u,v} \left[ \int_{\mathcal{X}} u(x) d\mathbb{P}(x) + \int_{\mathcal{Y}} v(y) d\mathbb{Q}(y) - \mathcal{F}^*(u \oplus v) \right], \tag{11}$$

where optimization is performed over $u, v \in \mathcal{C}(\mathcal{X}), \mathcal{C}(\mathcal{Y})$ which are called *potentials*. We use the dual form (11) to derive

$$\text{Cost}(\mathbb{P}, \mathbb{Q}) = \sup_{v} \left\{ \sup_{u} \left[ \int_{\mathcal{X}} u(x) d\mathbb{P}(x) - \mathcal{F}^*(u \oplus v) \right] + \int_{\mathcal{Y}} v(y) d\mathbb{Q}(y) \right\} = \tag{12}$$

$$\sup_{v} \left\{ \sup_{u} \left[ \int_{\mathcal{X}} u(x) d\mathbb{P}(x) - \sup_{\pi} \left( \int_{\mathcal{X} \times \mathcal{Y}} (u \oplus v) d\pi(x, y) - \mathcal{F}(\pi) \right) \right] + \int_{\mathcal{Y}} v(y) d\mathbb{Q}(y) \right\} = \tag{13}$$

$$\sup_{v} \left\{ \sup_{u} \left[ \int_{\mathcal{X}} u(x) d\mathbb{P}(x) + \inf_{\pi} \left( \mathcal{F}(\pi) - \int_{\mathcal{X} \times \mathcal{Y}} (u \oplus v) d\pi(x, y) \right) \right] + \int_{\mathcal{Y}} v(y) d\mathbb{Q}(y) \right\} = \tag{14}$$

$$\sup_{v} \left\{ \sup_{u} \inf_{\pi} \left( \mathcal{F}(\pi) - \int_{\mathcal{X}} u(x) d(\pi - \mathbb{P})(x) - \int_{\mathcal{Y}} v(y) d\pi(y) \right) + \int_{\mathcal{Y}} v(y) d\mathbb{Q}(y) \right\} \leq \tag{15}$$

$$\sup_{v} \left\{ \sup_{u} \inf_{\pi \in \Pi(\mathbb{P})} \left( \mathcal{F}(\pi) - \int_{\mathcal{X}} u(x) d(\pi - \mathbb{P})(x) - \int_{\mathcal{Y}} v(y) d\pi(y) \right) + \int_{\mathcal{Y}} v(y) d\mathbb{Q}(y) \right\} = \tag{16}$$

$$\sup_{v} \left\{ \sup_{u} \inf_{\pi \in \Pi(\mathbb{P})} \left( \mathcal{F}(\pi) - \int_{\mathcal{Y}} v(y) d\pi(y) \right) + \int_{\mathcal{Y}} v(y) d\mathbb{Q}(y) \right\} = \tag{17}$$

$$\sup_{v} \left\{ \inf_{\pi \in \Pi(\mathbb{P})} \left( \mathcal{F}(\pi) - \int_{\mathcal{Y}} v(y) d\pi(y) \right) + \int_{\mathcal{Y}} v(y) d\mathbb{Q}(y) \right\} \leq \tag{18}$$

$$\sup_{v} \left\{ \mathcal{F}(\pi^*) - \int_{\mathcal{Y}} v(y) \underbrace{d\pi^*(y)}_{d\mathbb{Q}(y)} + \int_{\mathcal{Y}} v(y) d\mathbb{Q}(y) \right\} = \mathcal{F}(\pi^*) = \text{Cost}(\mathbb{P}, \mathbb{Q}). \tag{19}$$

In line (12), we group the terms involving the potential $u$. In line (13), we express the conjugate functional $\mathcal{F}^*$ by using its definition. In the transition to line (14), we replace $\inf_\pi$ operator with the equivalent $\sup_\pi$ operator with the changed sign. In transition to (15), we put the term $\int_{\mathcal{X}} u(x) d\mathbb{P}(x)$ under the $\inf_\pi$ operator; we use definition $(u \oplus v)(x, y) = u(x) + v(y)$ to split the integral over $\pi(x, y)$ into two separate integrals over $\pi(x)$ and $\pi(y)$ respectively. In transition to (16), we restrict the inner $\inf_\pi$ to probability distributions $\pi \in \Pi(\mathbb{P})$ which have $\mathbb{P}$ as the first marginal, i.e. $d\pi(x) = d\mathbb{P}(x)$. This provides an upper bound on (15), in particular, all $u$-dependent terms vanish, see (17). As a result, we remove the $\sup_u$ operator in line (18). In transition to line (19) we substitute an optimal plan $\pi^* \in \Pi(\mathbb{P}, \mathbb{Q}) \subset \Pi(\mathbb{Q})$ to upper bound (18). Since $\text{Cost}(\mathbb{P}, \mathbb{Q})$ turns to be both an upper bound

(19) and a lower bound (12) for (18), we conclude that:

$$\text{Cost}(\mathbb{P}, \mathbb{Q}) = \sup_v \inf_{\pi \in \Pi(\mathbb{P})} \left\{ \mathcal{F}(\pi) - \int_{\mathcal{Y}} v(y)d\pi(y) + \int_{\mathcal{Y}} v(y)d\mathbb{Q}(y) \right\} \stackrel{\text{def}}{=} \sup_v \inf_{\pi \in \Pi(\mathbb{P})} \mathcal{L}_p(v, \pi). \quad (20)$$

The desired equation (5) is obtained by replacing plans $\pi \in \Pi(\mathbb{P})$ in (20) with their stochastic map representations $T_\pi$, see the first paragraph of §4.1 $\qquad\square$

*Proof of Theorem 2.* Assume that $T^* \notin \arg\inf_T \mathcal{L}(v^*, T)$. This yields

$$\mathcal{L}(v^*, T^*) > \inf_T \mathcal{L}(v^*, T) = \text{Cost}(\mathbb{P}, \mathbb{Q}).$$

On the other hand, by substituting $T^*$ with the optimal OT plan $\pi_{T^*}$ we write:

$$\mathcal{L}(v^*, T^*) = \mathcal{F}(\pi_{T^*}) - \int_{\mathcal{Y}} v^*(y) \underbrace{d\pi_{T^*}(y)}_{d\mathbb{Q}(y)} + \int_{\mathcal{X}} v^*(y)d\mathbb{Q}(y) = \mathcal{F}(\pi_{T^*}) = \text{Cost}(\mathbb{P}, \mathbb{Q}).$$

which is a contradiction. Thus, the assumption is wrong and (6) holds.

Let $\mathcal{F}$ be strictly convex, i.e. (4) permits a unique OT plan $\pi^*$. From the first part of the theorem, it holds that the corresponding stochastic map $T_{\pi^*}$ solves (6), i.e., $T_{\pi^*} \in \arg\inf_T \mathcal{L}(v^*, T)$. Consequently, $\pi^* \in \inf_{\pi \in \Pi(\mathbb{P})} \mathcal{L}_p(v^*, \pi)$, see (20). Let $T \in \arg\inf_T \mathcal{L}(v^*, T)$ be another stochastic map solving (6). Similarly, $\pi_T \in \inf_{\pi \in \Pi(\mathbb{P})} \mathcal{L}_p(v^*, \pi)$. Since $\mathcal{F}$ is strictly convex, then $\mathcal{L}_p(v^*, \pi)$ is also strictly convex as a functional of $\pi$. The latter immediately yields that $\pi^* = \pi_T$, i.e, $T$ is a stochastic OT map. Combining this conclusion with the first part of the theorem, we derive:

$$T^* \in \arg\inf_T \mathcal{L}(v^*, T) \Leftrightarrow \pi_{T^*} \text{ is OT plan,}$$

which finishes the proof. $\qquad\square$

**Definition 2** (Strongly convex functional w.r.t. metric $\rho(\cdot, \cdot)$). *Let $\mathcal{F} : \mathcal{M}(\mathcal{X} \times \mathcal{Y}) \to \mathbb{R} \cup \{+\infty\}$ be a convex lower semi-continuous functional. Let $\mathcal{U} \subset \mathcal{P}(\mathcal{X} \times \mathcal{Y}) \subset \mathcal{M}(\mathcal{X} \times \mathcal{Y})$ be a convex subset such that $\exists \pi \in \mathcal{U} : \mathcal{F}(\pi) < +\infty$. Functional $\mathcal{F}$ is called $\beta$-strongly convex on $\mathcal{U}$ w.r.t. metric $\rho(\cdot, \cdot)$ if $\forall \pi_1, \pi_2 \in \mathcal{U}, \forall \alpha \in [0, 1]$ it holds:*

$$\mathcal{F}(\alpha\pi_1 + (1 - \alpha)\pi_2) \leq \alpha\mathcal{F}(\pi_1) + (1 - \alpha)\mathcal{F}(\pi_2) - \frac{\beta}{2}\alpha(1 - \alpha)\rho^2(\pi_1, \pi_2). \quad (21)$$

**Lemma 1** (Property of minimizers of strongly convex cost functionals). *Consider a lower-semicontinuous $\beta$-strongly convex in metric $\rho(\cdot, \cdot)$ on $\mathcal{U} \subset \mathcal{P}(\mathcal{X} \times \mathcal{Y})$ functional $\mathcal{F}$. Assume that $\pi^* \in \mathcal{U}$ satisfies $\mathcal{F}(\pi^*) = \inf_{\pi \in \mathcal{U}} \mathcal{F}(\pi)$. Then $\forall \pi \in \mathcal{U}$ it holds:*

$$\mathcal{F}(\pi^*) \leq \mathcal{F}(\pi) - \frac{\beta}{2}\rho^2(\pi^*, \pi). \quad (22)$$

*Proof of Lemma 1.* We substitute $\pi_1 = \pi^*$, $\pi_2 = \pi$ to formula (21) and fix $\alpha \in [0, 1]$. We obtain

$$\mathcal{F}(\alpha\pi^* + (1 - \alpha)\pi) \leq \alpha\mathcal{F}(\pi^*) + (1 - \alpha)\mathcal{F}(\pi) - \frac{\beta}{2}\alpha(1 - \alpha)\rho^2(\pi^*, \pi) \Longleftrightarrow$$

$$\underbrace{\mathcal{F}(\alpha\pi^* + (1 - \alpha)\pi)}_{\geq \inf_{\pi' \in \mathcal{U}} \mathcal{F}(\pi') = \mathcal{F}(\pi^*)} -\alpha\mathcal{F}(\pi^*) \leq (1 - \alpha)\mathcal{F}(\pi) - \frac{\beta}{2}\alpha(1 - \alpha)\rho^2(\pi^*, \pi) \Longrightarrow$$

$$(1 - \alpha)\mathcal{F}(\pi^*) \leq (1 - \alpha)\mathcal{F}(\pi) - \frac{\beta}{2}\alpha(1 - \alpha)\rho^2(\pi^*, \pi) \Longleftrightarrow$$

$$\mathcal{F}(\pi^*) \leq \mathcal{F}(\pi) - \frac{\beta}{2}\alpha\rho^2(\pi^*, \pi). \quad (23)$$

Taking the limit $\alpha \to 1-$ in inequality (23), we obtain (22). $\qquad\square$

**Theorem 5** (Error analysis via duality gaps)**.** *In the conditions of Theorem 3, consider the duality gaps for an approximate solution* $(\hat{v}, \hat{\pi}) \in \mathcal{C}(\mathcal{Y}) \times \Pi(\mathbb{P})$ *of* (20)*:*

$$\epsilon_1(\hat{v}, \hat{\pi}) \stackrel{\text{def}}{=} \mathcal{L}_p(\hat{v}, \hat{\pi}) - \inf_{\pi \in \Pi(\mathbb{P})} \mathcal{L}_p(\hat{v}, \pi), \quad (24) \quad \epsilon_2(\hat{v}) \stackrel{\text{def}}{=} \sup_v \inf_{\pi \in \Pi(\mathbb{P})} \mathcal{L}_p(v, \pi) - \inf_{\pi \in \Pi(\mathbb{P})} \mathcal{L}_p(\hat{v}, \pi), \quad (25)$$

*which are the errors of solving the outer* $\sup_v$ *and inner* $\inf_\pi$ *problems in* (20)*, respectively. Then for the OT plan* $\pi^*$ *in* (4) *between* $\mathbb{P}$ *and* $\mathbb{Q}$ *the following inequality holds*

$$\rho(\hat{\pi}, \pi^*) \leq \sqrt{\frac{2}{\beta}} \left( \sqrt{\epsilon_1(\hat{v}, \hat{\pi})} + \sqrt{\epsilon_2(\hat{v})} \right), \quad (26)$$

*i.e., the sum of the roots of duality gaps upper bounds the error of the plan* $\hat{\pi}$ *w.r.t.* $\pi^*$ *in* $\rho(\cdot, \cdot)$*.*

*Proof of Theorem 5.* Given a potential $v \in \mathcal{C}(\mathcal{Y})$, we define functional $\mathcal{V}_v : \Pi(\mathbb{P}) \to \mathbb{R} \cup \{+\infty\}$:

$$\mathcal{V}_v(\pi) \stackrel{\text{def}}{=} \mathcal{F}(\pi) - \int_{\mathcal{Y}} v(y) d\pi(y). \quad (27)$$

Since the term $\int_{\mathcal{Y}} v(y) d\pi(y)$ is linear w.r.t. $\pi$, the $\beta$-strong convexity of $\mathcal{F}$ implies $\beta$-strong convexity of $\mathcal{V}_v$. Moreover, since $\mathcal{V}_v$ is lower semi-continuous and $\Pi(\mathbb{P})$ is compact (w.r.t. weak-$*$ topology), it follows from the Weierstrass theorem (Santambrogio, 2015, Box 1.1) that

$$\exists \pi^v \in \Pi(\mathbb{P}) : \mathcal{V}_v(\pi^v) = \inf_{\pi \in \Pi(\mathbb{P})} \mathcal{V}_v(\pi); \quad (28)$$

i.e. the infimum of $\mathcal{V}_v(\pi)$ is attained. Note that $\pi^v$ minimizes the functional $\pi \mapsto \mathcal{L}(v, \pi)$ as well since $\mathcal{L}(v, \pi) = \mathcal{V}_v(\pi) + \text{Const}(v)$. Therefore, the duality gaps (24), (25) permit the following reformulation:

$$\epsilon_1(\hat{v}, \hat{\pi}) = \mathcal{L}_p(\hat{v}, \hat{\pi}) - \mathcal{L}_p(\hat{v}, \pi^{\hat{v}}), \quad (29)$$

$$\epsilon_2(\hat{v}) = \mathcal{L}_p(v^*, \pi^*) - \mathcal{L}_p(\hat{v}, \pi^{\hat{v}}); \quad (30)$$

where $\pi^{\hat{v}}$ is a minimizer (28) for $v = \hat{v}$. Consider expression (29):

$$\epsilon_1(\hat{v}, \hat{\pi}) = \mathcal{L}_p(\hat{v}, \hat{\pi}) - \mathcal{L}_p(\hat{v}, \pi^{\hat{v}}) \stackrel{\text{Lemma 1}}{\geq} \frac{\beta}{2} \rho^2(\hat{\pi}, \pi^{\hat{v}}) \Longrightarrow$$

$$\sqrt{\frac{2}{\beta} \epsilon_1(\hat{v}, \hat{\pi})} \geq \rho(\hat{\pi}, \pi^{\hat{v}}). \quad (31)$$

Consider expression (30):

$$\epsilon_2(\hat{v}) = \mathcal{L}_p(v^*, \pi^*) - \mathcal{L}_p(\hat{v}, \pi^{\hat{v}}) =$$

$$\mathcal{F}(\pi^*) - \int_{\mathcal{Y}} v^*(y) d(\pi^* - \mathbb{Q})(y) - \mathcal{F}(\pi^{\hat{v}}) + \int_{\mathcal{Y}} \hat{v}(y) d(\pi^{\hat{v}} - \mathbb{Q})(y) =$$

$$\mathcal{F}(\pi^*) - \int_{\mathcal{Y}} \hat{v}(y) d(\pi^* - \mathbb{Q})(y) + \int_{\mathcal{Y}} \{\hat{v}(y) - v^*(y)\} d(\pi^* - \mathbb{Q})(y) - \mathcal{F}(\pi^{\hat{v}}) + \int_{\mathcal{Y}} \hat{v}(y) d(\pi^{\hat{v}} - \mathbb{Q})(y) = \quad (32)$$

$$\underbrace{\mathcal{F}(\pi^*) - \int_{\mathcal{Y}} \hat{v}(y) d\pi^*(y)}_{= \mathcal{V}_{\hat{v}}(\pi^*)} + \underbrace{\int_{\mathcal{Y}} \{\hat{v}(y) - v^*(y)\} d(\pi^* - \mathbb{Q})(y)}_{= 0, \text{ since } d\pi^*(y) = d\mathbb{Q}(y)} \underbrace{- \mathcal{F}(\pi^{\hat{v}}) + \int_{\mathcal{Y}} \hat{v}(y) d\pi^{\hat{v}}(y)}_{= -\mathcal{V}_{\hat{v}}(\pi^{\hat{v}})} =$$

$$\mathcal{V}_{\hat{v}}(\pi^*) - \mathcal{V}_{\hat{v}}(\pi^{\hat{v}}) \stackrel{\text{Lemma 1}}{\geq} \frac{\beta}{2} \rho^2(\pi^*, \pi^{\hat{v}}) \Longrightarrow$$

$$\sqrt{\frac{2}{\beta} \epsilon_2(\hat{v})} \geq \rho(\pi^*, \pi^{\hat{v}}); \quad (33)$$

where in line (32) we add and subtract $\int_{\mathcal{Y}} \hat{v}(y) d(\pi^* - \mathbb{Q})(y)$.

The triangle inequality $\rho(\pi^*, \hat{\pi}) \leq \rho(\pi^*, \pi^{\hat{v}}) + \rho(\hat{\pi}, \pi^{\hat{v}}) = \sqrt{\frac{2}{\beta}} \left( \sqrt{\epsilon_1(\hat{v}, \hat{\pi})} + \sqrt{\epsilon_2(\hat{v})} \right)$ for norm $\rho(\cdot, \cdot)$ finishes the proof. $\square$

*Proof of Theorem 3.* The statement directly follows from Theorem 5 by substituting plans $\pi \in \Pi(\mathbb{P})$ with their stochastic map representations $T_\pi$, see the first paragraph of §4.1. $\square$

### A.2 Proofs of Results of §5.1

*Proof of Theorem 4.* First, we prove that $\mathcal{F} = \mathcal{F}_\mathrm{G}$ it is *-separately increasing. For $\pi \in \mathcal{M}(\mathcal{X} \times \mathcal{Y}) \setminus \Pi(\mathbb{P})$ it holds that $\mathcal{F}(\pi) = +\infty$. Consequently,

$$\int_{\mathcal{X} \times \mathcal{Y}} c(x, y) d\pi(x, y) - \mathcal{F}(\pi) = \int_{\mathcal{X} \times \mathcal{Y}} \big( u(x) + v(y) \big) d\pi(x, y) - \mathcal{F}(\pi) = -\infty. \qquad (34)$$

When $\pi \in \Pi(\mathbb{P})$ it holds that $\pi$ is a probability distribution. We integrate $u(x) + v(y) \leq c(x, y)$ w.r.t. $\pi$, subtract $\mathcal{F}(\pi)$ and obtain

$$\int_{\mathcal{X} \times \mathcal{Y}} c(x, y) d\pi(x, y) - \mathcal{F}(\pi) \geq \int_{\mathcal{X} \times \mathcal{Y}} \big( u(x) + v(y) \big) d\pi(x, y) - \mathcal{F}(\pi). \qquad (35)$$

By taking the $\sup$ of (34) and (35) w.r.t. $\pi \in \mathcal{M}(\mathcal{X} \times \mathcal{Y})$, we obtain $\mathcal{F}^*(c) \geq \mathcal{F}^*(u \oplus v)$.[1]

Next, we prove that $\mathcal{F}$ is convex. We prove that every term $\mathcal{E}^2\big(T_\pi \sharp(\mathbb{P}_n \times \mathbb{S}), \mathbb{Q}_n\big)$ is convex in $\pi$.

**First**, we show that $\pi \mapsto f_n(\pi) \overset{def}{=} T_\pi \sharp(\mathbb{P}_n \times \mathbb{S})$ is *linear* in $\pi \in \Pi(\mathbb{P})$.

Pick any $\pi_1, \pi_2, \pi_3 \in \Pi(\mathbb{P})$ which lie on the same line. Without loss of generality we assume that $\pi_3 \in [\pi_1, \pi_2]$, i.e., $\pi_3 = \alpha\pi_1 + (1 - \alpha)\pi_2$ for some $\alpha \in [0, 1]$. We need to show that

$$f_n(\pi_3) = \alpha f_n(\pi_1) + (1 - \alpha)f_n(\pi_2). \qquad (36)$$

In what follows, for a random variable $U$ we denote its distribution by $\mathrm{Law}(U)$.

The first marginal distribution of each $\pi_i$ is $\mathbb{P}$. From the glueing lemma (Villani, 2008, §1) it follows that there exists a triplet of (dependent) random variables $(X, Y_1, Y_2)$ such that $\mathrm{Law}(X, Y_i) = \pi_i$ for $i = 1, 2$. We define $Y_3 = Y_r$, where $r$ is an *independent* random variable which takes values in $\{1, 2\}$ with probabilities $\{\alpha, 1 - \alpha\}$. From the construction of $Y_3$ it follows that $\mathrm{Law}(X, Y_3)$ is a mixture of $\mathrm{Law}(X, Y_1) = \pi_1$ and $\mathrm{Law}(X, Y_2) = \pi_2$ with weights $\alpha$ and $1 - \alpha$. Thus, $\mathrm{Law}(X, Y_3) = \alpha\pi_1 + (1 - \alpha)\pi_2 = \pi_3$. We conclude that $\mathrm{Law}(Y_3|X = x) = \pi_3(\cdot|x)$ for $\mathbb{P}$-almost all $x \in \mathcal{X}$ (recall that $\mathrm{Law}(X) = \mathbb{P}$). On the other hand, again by the construction, the conditional $\mathrm{Law}(Y_3|X = x)$ is a mixture of $\mathrm{Law}(Y_1|X = x) = \pi_1(\cdot|x)$ and $\mathrm{Law}(Y_2|X = x) = \pi_2(\cdot|x)$ with weights $\alpha$ and $1 - \alpha$. Thus, $\pi_3(\cdot|x) = \alpha\pi_1(\cdot|x) + (1 - \alpha)\pi_2(\cdot|x)$ holds true for $\mathbb{P}$-almost all $x \in \mathcal{X}$.

Consider independent random variables $X_n \sim \mathbb{P}_n$ and $Z \sim \mathbb{S}$. From the definition of $T_{\pi_i}$ we conclude that $\mathrm{Law}\big(T_{\pi_i}(x, Z)\big) = \pi_i(\cdot|x)$ for $\mathbb{P}$-almost all $x \in \mathcal{X}$ and, since $\mathbb{P}_n$ is a component of $\mathbb{P}$, for $\mathbb{P}_n$-almost all $x \in \mathcal{X}$ as well. As a result, we define $T_i = T_{\pi_i}(X_n, Z)$ and derive

$$\mathrm{Law}(T_3|X_n = x) = \pi_3(\cdot|x) = \alpha\pi_1(\cdot|x) + (1 - \alpha)\pi_2(\cdot|x) =$$
$$\alpha\mathrm{Law}(T_1|X_n = x) + (1 - \alpha)\mathrm{Law}(T_2|X_n = x)$$

for $\mathbb{P}_n$-almost all $x \in \mathcal{X}$. Thus, $\mathrm{Law}(X_n, T_3)$ is also a mixture of $\mathrm{Law}(X_n, T_1)$ and $\mathrm{Law}(X_n, T_2)$ with weights $\alpha$ and $1 - \alpha$. In particular, $\mathrm{Law}(T_3) = \alpha\mathrm{Law}(T_1) + (1 - \alpha)\mathrm{Law}(T_2)$. We note that $\mathrm{Law}(T_i) = f_n(\pi_i)$ by the definition of $f_n$ and obtain (36).

**Second**, we highlight that for every $\nu \in \mathcal{P}(\mathcal{Y})$, the functional $\mathcal{P}(\mathcal{Y}) \ni \mu \to \mathcal{E}^2(\mu, \nu)$ is convex in $\mu$. Indeed, $\mathcal{E}^2$ is a particular case of (the square of) Maximum Mean Discrepancy (MMD, (Sejdinovic et al., 2013)). Therefore, there exists a Hilbert space $\mathcal{H}$ and a function $\phi : \mathcal{Y} \to \mathcal{H}$ (feature map), such that

$$\mathcal{E}^2(\mu, \nu) = \left\| \int_{\mathcal{Y}} \phi(y) d\mu(y) - \int_{\mathcal{Y}} \phi(y) d\nu(y) \right\|_{\mathcal{H}}^2.$$

Since the kernel mean embedding $\mu \mapsto \int_{\mathcal{Y}} \phi(y) d\mu(y)$ is linear in $\mu$ and $\| \cdot \|_{\mathcal{H}}^2$ is convex, we conclude that $\mathcal{E}^2(\mu, \nu)$ is convex in $\mu$. To finish this part of the proof, it remains to combine the fact that $\pi \mapsto T_\pi \sharp(\mathbb{P}_n \times \mathbb{S})$ is linear and $\mathcal{E}^2(\cdot, \mathbb{Q}_n)$ is convex in the first argument.

**Third**, we note that the lower semi-continuity of $\mathcal{F}(\pi)$ in $\Pi(\mathbb{P})$ follows from the lower semi-continuity of the Energy distance ($\mathcal{E}^2$) terms in (7). That is, it suffices to show that $\mathcal{E}^2$ defined in equation (8)

---

[1]The proof is generic and works for any functional which equals $+\infty$ outside $\pi \in \mathcal{P}(\mathcal{X} \times \mathcal{Y})$.

is indeed lower semi-continuous in the first argument $\mathbb{Q}_1$. In (8), there are two terms depending on $\mathbb{Q}_1$. The term $\mathbb{E}\|X_1 - X_1'\|_2 = \int_{\mathcal{X}} \left[ \int_{\mathcal{X}} \|x_1 - x_2\|_2 d\mathbb{Q}_2(x_2) \right] d\mathbb{Q}_1(x_1)$ is linear in $\mathbb{Q}_1$. It is just the expectation of a continuous function w.r.t. $\mathbb{Q}_1$. Hence it is lower semi-continuous by the definition of the lower semi-continuity. Here we also use the fact that $\mathcal{Y}$ is compact. The other term $-\frac{1}{2}\mathbb{E}\|X_1 - X_1'\|_2 = -\frac{1}{2}\int_{\mathcal{X}\times\mathcal{X}} \|x_2 - x_2'\|_2 d(\mathbb{Q}_1 \times \mathbb{Q}_1)(x_1, x_2)$ is a quadratic term in $\mathbb{Q}_1$. This term can be viewed as the interaction energy (Santambrogio, 2015, §7) between particles in $\mathbb{Q}_1$ with the interaction function $W(x_1, x_1') \overset{def}{=} -\|x_1 - x_1'\|_2$. Thanks to the compactness of $\mathcal{Y}$, it is also lower semi-continuous in $\mathbb{Q}_1$, see (Santambrogio, 2015, Proposition 7.2) for the proof. $\qquad\square$

*Proof of Proposition 1.* Direct calculation of the expectation of (9) yields the value

$$\mathbb{E}\|Y - T(X, Z)\|_2 - \frac{1}{2}\mathbb{E}\|T(X, Z) - T(X', Z')\|_2 =$$

$$\mathbb{E}\|Y - T(X, Z)\| - \frac{1}{2}\mathbb{E}\|T(X, Z) - T(X', Z')\|_2 - \frac{1}{2}\mathbb{E}\|Y - Y'\|_2 + \frac{1}{2}\mathbb{E}\|Y - Y'\|_2 =$$

$$\mathcal{E}^2\big(T\sharp(\mathbb{P}_n \times \mathbb{S}), \mathbb{Q}_n\big) + \frac{1}{2}\mathbb{E}\|Y - Y'\|_2, \quad (37)$$

where $Y, Y' \sim \mathbb{Q}_n$ and $(X, Z), (X', Z') \sim (\mathbb{P}_n \times \mathbb{S})$ are independent random variables. It remains to note that $\frac{1}{2}\mathbb{E}\|Y - Y'\|_2$ is a $T$-independent constant. $\qquad\square$

## B  ALGORITHM FOR GENERAL COST FUNCTIONALS

In this section, we present the procedure to optimize (5) for general cost functionals $\mathcal{F}$. In practice, one may utilize neural networks $T_\theta : \mathbb{R}^D \times \mathbb{R}^S \to \mathbb{R}^D$ and $v_\omega : \mathbb{R}^D \to \mathbb{R}$ to parameterize $T$ and $v_\omega$, correspondingly, to solve the problem (5). One may train them with stochastic gradient ascent-descent (SGAD) using random batches from $\mathbb{P}, \mathbb{Q}, \mathbb{S}$. The procedure is summarized in Algorithm 2.

---

**Algorithm 2:** Neural optimal transport for general cost functionals

**Input**  : Distributions $\mathbb{P}, \mathbb{Q}, \mathbb{S}$ accessible by samples; mapping network $T_\theta : \mathbb{R}^P \times \mathbb{R}^S \to \mathbb{R}^Q$;
potential network $v_\omega : \mathbb{R}^Q \to \mathbb{R}$; number of inner iterations $K_T$; empirical estimator
$\widehat{\mathcal{F}}\big(X, T(X, Z)\big)$ for cost $\widetilde{\mathcal{F}}(T)$;

**Output** : Learned stochastic OT map $T_\theta$ representing an OT plan between distributions $\mathbb{P}, \mathbb{Q}$;

**repeat**
> Sample batches $Y \sim \mathbb{Q}$, $X \sim \mathbb{P}$ and for each $x \in X$ sample batch $Z[x] \sim \mathbb{S}$;
> $\mathcal{L}_v \leftarrow \sum\limits_{x \in X} \sum\limits_{z \in Z[x]} \frac{v_\omega(T_\theta(x,z))}{|X| \cdot |Z[x]|} - \sum\limits_{y \in Y} \frac{v_\omega(y)}{|Y|}$;
> Update $\omega$ by using $\frac{\partial \mathcal{L}_v}{\partial \omega}$;
> **for** $k_T = 1, 2, \ldots, K_T$ **do**
>> Sample batch $X \sim \mathbb{P}$ and for each $x \in X$ sample batch $Z[x] \sim \mathbb{S}$;
>> $\mathcal{L}_T \leftarrow \widehat{\mathcal{F}}(X, T_\theta(X, Z)) - \sum\limits_{x \in X} \sum\limits_{z \in Z[x]} \frac{v_\omega(T_\theta(x,z))}{|X| \cdot |Z[x]|}$;
>> Update $\theta$ by using $\frac{\partial \mathcal{L}_T}{\partial \theta}$;

**until** not converged;

---

Algorithm 2 requires an empirical estimator $\widehat{\mathcal{F}}$ for $\widetilde{\mathcal{F}}(T)$. Providing such an estimator might be non-trivial for general $\mathcal{F}$. If $\mathcal{F}(\pi) = \int_{\mathcal{X}} C(x, \pi(\cdot|x)) d\mathbb{P}(x)$, i.e., the cost is weak (2), one may use the following *unbiased* Monte-Carlo estimator: $\widehat{\mathcal{F}}\big(X, T(X, Z)\big) \overset{def}{=} |X|^{-1} \sum_{x \in X} \widehat{C}\big(x, T(x, Z[x])\big)$, where $\widehat{C}$ is the respective estimator for the weak cost $C$ and $Z[x]$ denotes a random batch of latent vectors $z \sim \mathbb{S}$ for a given $x \in \mathcal{X}$. For classic costs and the $\gamma$-weak quadratic cost, the estimator $\widehat{C}$ is given by (Korotin et al., 2023b, Eq. 18 and 19) and Algorithm 2 for general OT 4 reduces to the neural OT algorithm (Korotin et al., 2023b, Algorithm 1) for weak (2) or classic (1) OT. Unlike the predecessor, the algorithm is suitable for *general* OT formulation (4). In §5.1, we propose a cost functional $\mathcal{F}_G$ to solve the class-guided dataset transfer task (Algorithm 1). In §5.2, we propose the functional $\mathcal{F}_S$ for the paired domain translataion (Algorithm 3).

## C  CLASS-GUIDED EXPERIMENTS

### C.1  TRAINING AND COMPARISON DETAILS

The code is written in `PyTorch` framework and publicly available at `https://github.com/machinestein/gnot`. On the image data, our method converges in 5–15 hours on a Tesla V100 (16 GB). We use WandB for babysitting the experiments (Biewald, 2020).

**Algorithm details**. In our Algorithm 1, we use Adam (Kingma & Ba, 2014) optimizer with $lr = 10^{-4}$ for both $T_\theta$ and $v_\omega$. The number of inner iterations for $T_\theta$ is $K_T = 10$. Doing preliminary experiments, we noted that it is sufficient to use small mini-batch sizes $K_X, K_Y, K_Z$ in (9). Therefore, we decided to average loss values over $K_B$ small independent mini-batches (each from class $n$ with probability $\alpha_n$) rather than use a single large batch from one class. This is done parallel with tensor operations.

**Dataset an hyperparameters.** We rescale the images to size 32×32 and normalize their channels to $[-1, 1]$. For the grayscale images, we repeat their channel 3 times and work with 3-channel images. We do not apply any augmentations to data. We use the default train-test splits for all the datasets.

We use WGAN-QC discriminator's ResNet architecture (He et al., 2016) for potential $v_\omega$. We use UNet[2] (Ronneberger et al., 2015) as the stochastic transport map $T_\theta(x, z)$. To condition it on $z$, we insert conditional instance normalization (CondIN) layers after each UNet's upscaling block[3]. We use CondIN from AugCycleGAN (Almahairi et al., 2018). In experiments, $z$ is the 128-dimensional standard Gaussian noise.

The batch size is $K_B = 32$, $K_X = K_Y = 2$, $K_Z = 2$ for training with $z$. When training without $z$, we use the original UNet without conditioning; the batch parameters are the same ($K_Z$ does not matter). Our method converges in $\approx 60k$ iterations of $v_\omega$.

For comparison in the image domain, we use the official implementations with the hyperparameters from the respective papers: AugCycleGAN[4] (Almahairi et al., 2018), MUNIT[5](Huang et al., 2018). For comparison with neural OT ($\mathbb{W}_2$, $\mathcal{W}_{2,\gamma}$), we use their publicly available code.[6]. For the stochastic maps $T(x, z)$, we only sampled one $z$ per $x$ during computing the metric, no averaging over $z$ was applied. The accuracy of the ResNet18 classifiers is 99.17 for the MNIST and 95.56 for the USPS datasets. For the KMNIST and MNITST, the accuracy's are 99.39 and 97.19, respectively.

**OTDD flow details.** As in our method, the number of labelled samples in each class is 10. We learn the OTDD flow between the labelled source dataset[7] and labelled target samples. Note the OTDD method does not use the unlabeled target samples. As the OTDD method does not produce out-of-sample estimates, we train UNet to map the source data to the data produced by the OTDD flow via regression. Then we compute the metrics on the test (FID, accuracy) for this mapping network.

**DOT details.** Input pre-processing was the same as in our method. We tested a variety of discrete OT solvers from Python Optimal Transport (POT) package (Flamary et al., 2021), including EMD, MappingTransport (Perrot et al., 2016) and SinkhornTransport with Laplacian and L2 regularization (Courty et al., 2016) from `ot.da` (Flamary et al., 2021). These methods are semi-supervised and can receive labels to construct a task-specific plan. As in our method, the number of labelled samples in each class is 10. For most of these methods, two tunable hyper-parameters are available: entropic and class regularization values. We evaluated a range of these values (1, 2, 5, 10, and 100). To assess the accuracy of the DOT solvers, we used the same oracle classifiers as in all the other cases. Empirically, we found that the Sinkhorn with Laplacian regularization and both regularization values equal to 5 achieves the best performance in most cases. Thus, to keep Table 1 simple, we report the test accuracy results only for this DOT approach. Additionally, we calculated its test FID (Table 2).

---

[2]`github.com/milesial/Pytorch-UNet`

[3]`github.com/kgkgzrtk/cUNet-Pytorch`

[4]`github.com/aalmah/augmented_cyclegan`

[5]`github.com/NVlabs/MUNIT`

[6]`https://github.com/iamalexkorotin/NeuralOptimalTransport`

[7]We use only 15k source samples since OTDD is computationally heavy (the authors use 2k samples).

## C.2 MOONS

The task is to map two balanced classes of moons (red and green) between $\mathbb{P}$ and $\mathbb{Q}$ (circles and crosses in Figure 5(a), respectively). The target distribution $\mathbb{Q}$ is $\mathbb{P}$ rotated by 90 degrees. The number of randomly picked labelled samples in each target moon is *10*. The maps learned by neural OT algorithm with the quadratic cost ($\mathbb{W}_2$, (Fan et al., 2023; Korotin et al., 2023b)) and **our** Algorithm 1 with functional $\mathcal{F}_G$ are given in Figures 5(c) and 5(d), respectively. In Figure 5(b) we show the *matching* performed by a *discrete* OT-SI algorithm which learns the transport cost with a neural net from a known classes' correspondence (Liu et al., 2020). As expected, the map for $\mathbb{W}_2$ does not preserve the classes (Figure 5(c)), while our map solves the task (Figure 5(d)). We use 500 train and 150 test samples for each moon. We use the fully-connected net with 2 ReLU hidden layers size of 128 for both $T_\theta$ and $v_\omega$. We train the model for 10k iterations of $v_\omega$ with $K_B = 32, K_X = K_Y = 2$ ($K_Z$ plays no role as we do not use $z$ here).

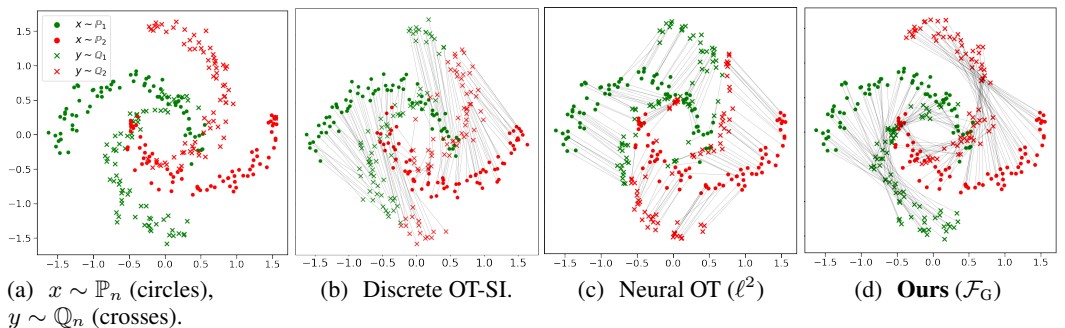

(a) $x \sim \mathbb{P}_n$ (circles), $y \sim \mathbb{Q}_n$ (crosses).   (b) Discrete OT-SI.   (c) Neural OT ($\ell^2$)   (d) **Ours** ($\mathcal{F}_G$)

Figure 5: The results of mapping two moons using OT with different cost functionals.

## C.3 GAUSSIANS MIXTURES.

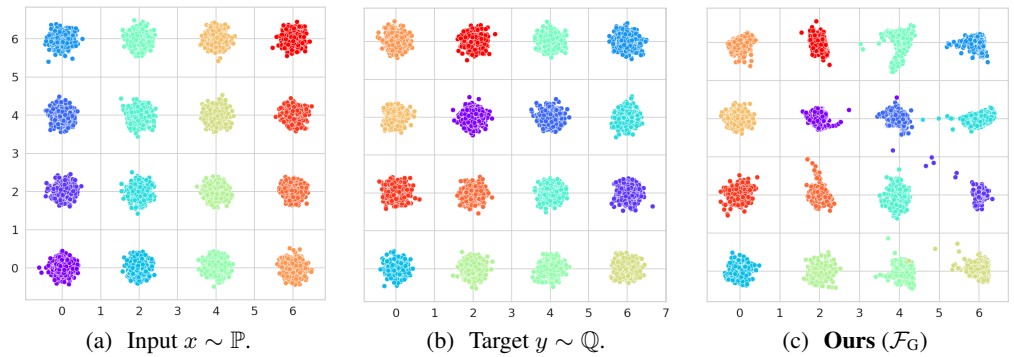

(a) Input $x \sim \mathbb{P}$.   (b) Target $y \sim \mathbb{Q}$.   (c) **Ours** ($\mathcal{F}_G$)

Figure 6: Illustration of the mapping between two Gaussian mixtures learned by our Algorithm 1.

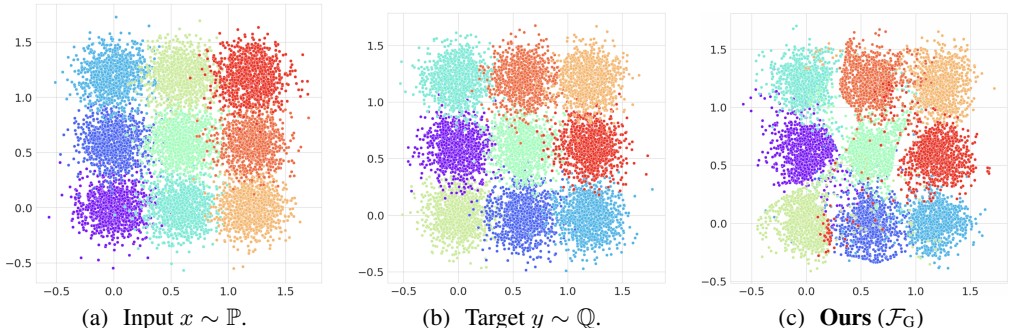

(a) Input $x \sim \mathbb{P}$.   (b) Target $y \sim \mathbb{Q}$.   (c) **Ours** ($\mathcal{F}_G$)

Figure 7: Illustration of the mapping between two Gaussian mixtures learned by our Algorithm 1 when the classes are overlapping.

In this additional experiment both $\mathbb{P}, \mathbb{Q}$ are balanced mixtures of 16 Gaussians, and each color denotes a unique class. The goal is to map Gaussians in $\mathbb{P}$ (Figure 6(a)) to respective Gaussians in $\mathbb{Q}$ which have the same color, see Figure 6(b). The result of our method (*10* known target labels per class) is given in Figure 6(c). It correctly maps the classes. Neural OT for the quadratic cost is not shown as it results in the *identity map* (the same image as Figure 6(a)) which is completely *mistaken* in classes. We use the fully connected network with 2 ReLU hidden layers size of 256 for both $T_\theta$ and $v_\omega$. There are 10000 train and 500 test samples in each Gaussian. We train the model for 10k iterations of $v_\omega$ with $K_B = 32, K_X = K_Y = 2$ ($K_Z$ plays no role here as well).

Using the same settings as in the previous experiment, we conducted additional tests involving overlapping classes. In this scenario, specific samples within one class are identical to samples from another class. To execute this experiment, we adjusted the Gaussian modes of each class to be closer to each other, as illustrated in Figure 7(a). The target classes are depicted in Figure 7(b). Our method handles this scenario, demonstrating the robustness of our model in handling overlapping classes. The visual representation can be observed in Figure 7(c).

## C.4 Experiments with other image datasets

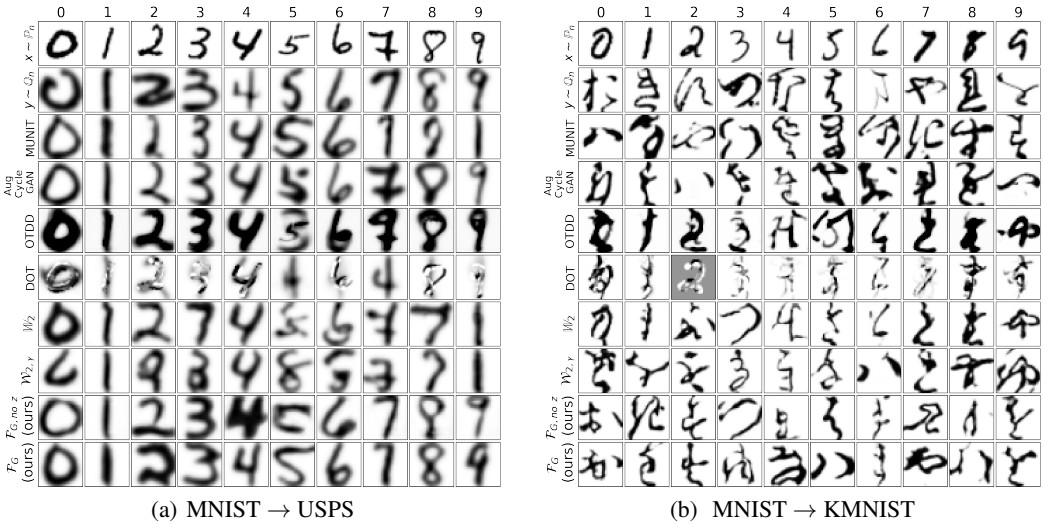

(a) MNIST → USPS          (b) MNIST → KMNIST

Figure 8: The results of mapping between two datasets.

| Datasets ($32 \times 32$) | Image-to-Image Translation | | Flows | Discrete OT | Neural Optimal Transport | | | |
|---|---|---|---|---|---|---|---|---|
| | MUNIT | Aug CycleGAN | OTDD | SinkhornLpL1 | $\mathbb{W}_2$ | $\mathcal{W}_{2,\gamma}$ | $\mathcal{F}_\mathrm{G}$, no $z$ [Ours] | $\mathcal{F}_\mathrm{G}$ [Ours] |
| MNIST → USPS | 97.95 | **98.2** | - | 83.26 | 38.77 | 37.0 | 95.27 | 94.62 |
| MNIST → KMNIST | 12.27 | 8.99 | 4.46 | 4.27 | 6.13 | 6.82 | **79.20** | 61.91 |

Table 3: Accuracy↑ of the maps learned by the translation methods in view.

| Datasets ($32 \times 32$) | MUNIT | Aug CycleGAN | OTDD | SinkhornLpL1 | $\mathbb{W}_2$ | $\mathcal{W}_{2,\gamma}$ | $\mathcal{F}_\mathrm{G}$, no $z$ [Ours] | $\mathcal{F}_\mathrm{G}$ [Ours] |
|---|---|---|---|---|---|---|---|---|
| MNIST → USPS | 6.86 | 22.74 | > 100 | 51.18 | 4.60 | 3.05 | 5.40 | **2.87** |
| MNIST → KMNIST | 8.81 | 62.19 | > 100 | 40.96 | 12.85 | **9.46** | 17.26 | 9.69 |

Table 4: FID↓ of the samples generated by the translation methods in view.

Here we test the case with different datasets, again in one case when the source and target domains are related, in the second case when they are not. We consider MNIST→USPS and MNIST→KMNIST. As in the main text, we are given only 10 labeled samples from the target dataset; the rest are unlabeled. The results are shown in Table 3, 4 and Figure 8.

In this case (Figure 8), GAN-based methods and our approach with our guided cost $\mathcal{F}_\mathrm{G}$ show high accuracy $\geq 90\%$. However, neural OT with classic and weak quadratic costs provides low accuracy (35-50%). We presume that this is because for these dataset pairs the ground truth OT map for

the (pixel-wise) quadratic cost simply does not preserve the class. This agrees with (Daniels et al., 2021, Figure 3) which tests an entropy-regularized quadratic cost in a similar MNIST→USPS setup. For our method with cost $\mathcal{F}_G$, The OTDD gradient flows method provides reasonable accuracy on MNIST→USPS. However, OTDD has a much higher FID than the other methods.

## C.5   ADDITIONAL VISUALIZATION

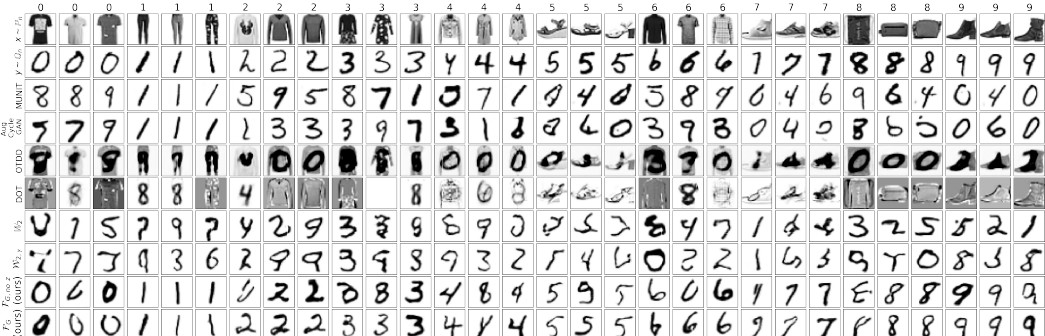

Figure 9: FMNIST→MNIST mapping results. Three input images per class are presented.

To further qualitative demonstrate that our model preserves the classes well, we provide an additional visualization of the learned maps. The same models were used as in Figure 6.1. The only difference is that in Figure 6.1 we show a single input and target per class, while here (Figure 9) we show three inputs and three outputs per class for the different methods.

## C.6   ADDITIONAL EXAMPLES OF STOCHASTIC MAPS

In this subsection, we provide additional examples of the learned stochastic map for $\mathcal{F}_G$ (with $z$). We consider all the image datasets from the main experiments (§6). The results are shown in Figure 11 and demonstrate that for a fixed $x$ and different $z$, our model generates diverse samples.

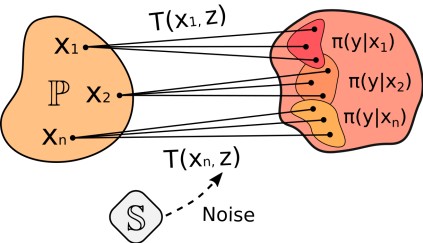

Figure 10: Implicit representation of $\pi \in \Pi(\mathbb{P})$ via function $T = T_\pi : \mathcal{X} \times \mathcal{Z} \to \mathcal{Y}$.

## C.7   ABLATION STUDY OF THE LATENT SPACE DIMENSION

In this subsection, we study the structure of the learned stochastic map for $\mathcal{F}_G$ with different latent space dimensions $Z$. We consider MNIST → USPS transfer task (10 classes). The results are shown in Figures 12, 13 and Table 5. As can be seen, our model performs comparably for different $Z$.

| Metrics | $Z = 1$ | $Z = 4$ | $Z = 8$ | $Z = 16$ | $Z = 32$ | $Z = 64$ |
|---|---|---|---|---|---|---|
| Accuracy | 86.96 | 93.48 | 91.82 | 92.08 | 92.25 | 92.95 |
| FID | 4.90 | 5.88 | 4.63 | 3.80 | 4.34 | 4.61 |

Table 5: Accuracy↑ and FID↓ of the stochastic maps MNIST → USPS learned by our translation method with different noise size $Z$.

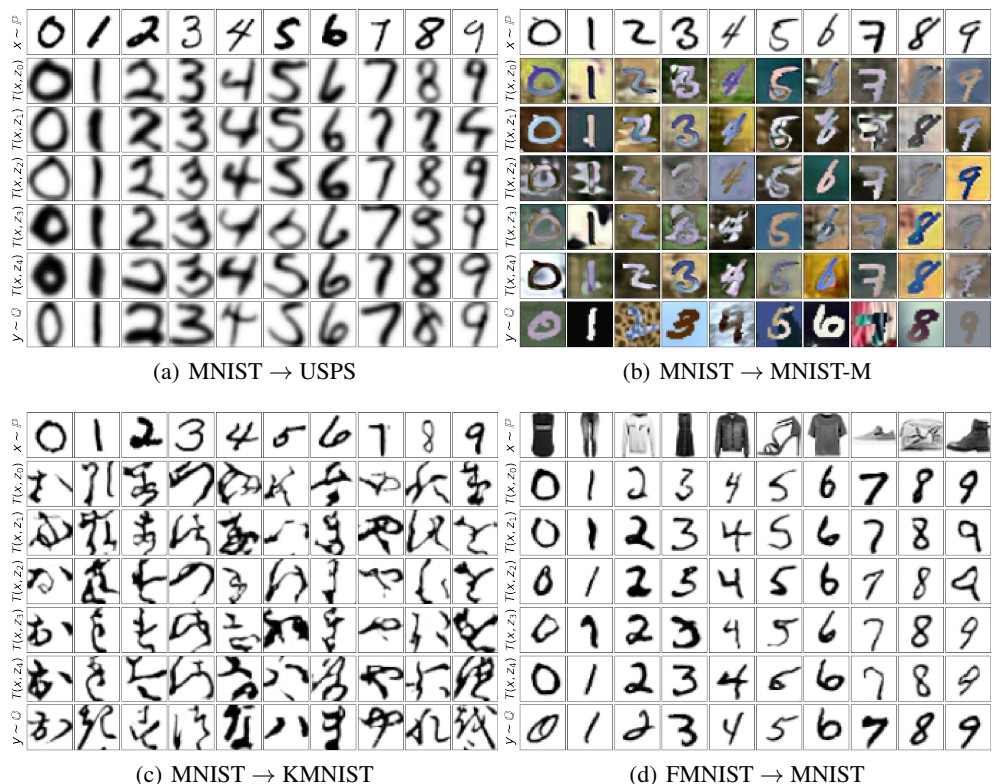

Figure 11: Stochastic transport maps $T_\theta(x, z)$ learned by our Algorithm 1. Additional examples.

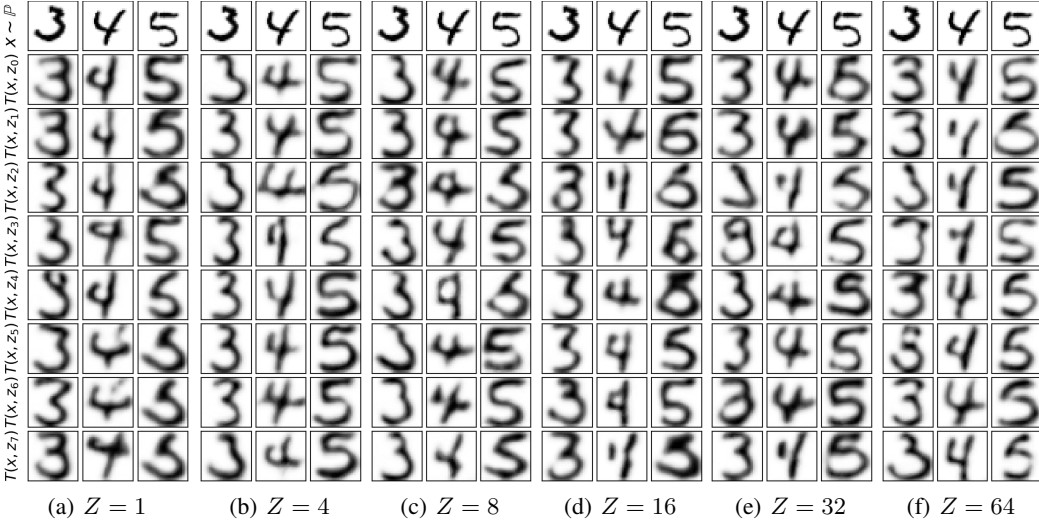

Figure 12: MNIST → USPS translation with functional $\mathcal{F}_G$ and varying $Z = 1, 4, 8, 16, 32, 64$.

## C.8 IMBALANCED CLASSES

In this subsection, we study the behaviour of the optimal map for $\mathcal{F}_G$ when the classes are imbalanced in input and target domains. Since our method learns a transport map from $\mathbb{P}$ to $\mathbb{Q}$, it should capture the class balance of the $\mathbb{Q}$ *regardless* of the class balance in $\mathbb{P}$. We check this below.

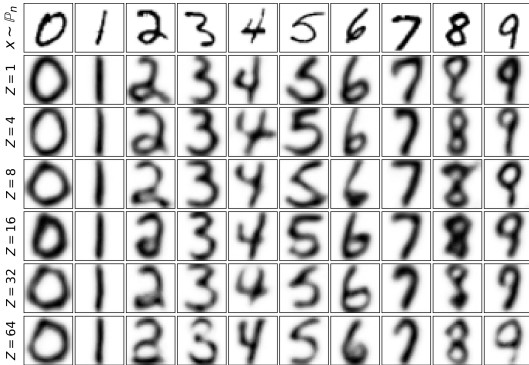

Figure 13: Stochastic transport maps $T_\theta(x, z)$ learned by our Algorithm 1 with different sizes of $Z$.

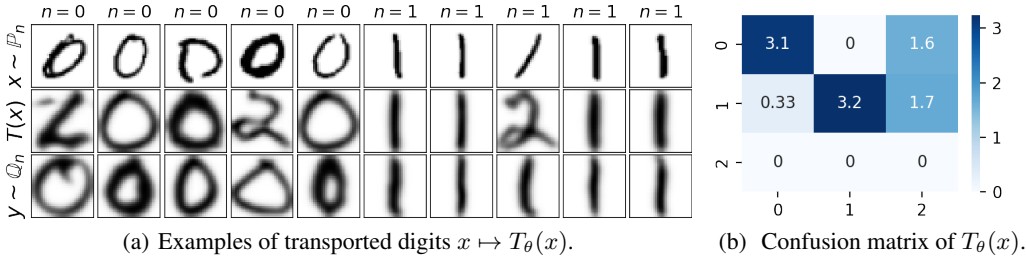

(a) Examples of transported digits $x \mapsto T_\theta(x)$.

(b) Confusion matrix of $T_\theta(x)$.

Figure 14: Imbalanced MNIST $\rightarrow$ USPS translation with functional $\mathcal{F}_G$ (deterministic, no $z$).

We consider MNIST $\rightarrow$ USPS datasets with $n = 3$ classes in MNIST and $n = 3$ classes in USPS. We assume that the class probabilities are $\alpha_1 = \alpha_2 = \frac{1}{2}$, $\alpha_3 = 0$ and $\beta_1 = \beta_2 = \beta_3 = \frac{1}{3}$. That is, there is no class 3 in the source dataset and it is not used anywhere during training. In turn, the target class 3 is not used when training $T_\theta$ but is used when training $f_\omega$. All the hyperparameters are the same as in the previous MNIST $\rightarrow$ USPS experiments with 10 known labels in target classes. The results are shown in Figures 14(a) and 15(a). We show deterministic (no $z$) and stochastic (with $z$) maps.

Our cost functional $\mathcal{F}_G$ stimulates the map to maximally preserve the input class. However, to transport $\mathbb{P}$ to $\mathbb{Q}$, the model *must* change the class balance. We show the confusion matrix for learned maps $T_\theta$ in Figures 14(b), 15(b). It illustrates that the model maximally preserves the input classes 0, 1 and uniformly distributes the input classes 0 and 1 into class 2, as suggested by our cost functional.

## C.9   ICNN-BASED DATASET TRANSFER

For completeness, we show the performance of ICNN-based method for the classic (1) quadratic transport cost $c(x, y) = \frac{1}{2}\|x - y\|^2$ on the dataset transfer task. We use the non-minimax version (Korotin et al., 2021a) of the ICNN-based method by (Makkuva et al., 2020). We employ the publicly available code and dense ICNN architectures from the Wasserstein-2 benchmark repository [8]. The batch size is $K_B = 32$, the total number of iterations is 100k, $lr = 3 \cdot 10^{-3}$, and the Adam optimizer is used. The datasets are preprocessed as in the other experiments, see Appendix C.1.

The qualitative results for MNIST$\rightarrow$USPS and FashionMNIST$\rightarrow$MNIST transfer are given in Figure 16. The results are reasonable in the first case (related domains). However, they are visually unpleasant in the second case (unrelated domains). This is expected as the second case is notably harder. More generally, as derived in the Wasserstein-2 benchmark (Korotin et al., 2021b), the ICNN models do not work well in the pixel space due to the poor expressiveness of ICNN architectures. The ICNN method achieved **18.8**% accuracy and $\gg$**100** FID in the FMNIST$\rightarrow$MNIST transfer, and **35.6**% and accuracy and **13.9** FID in the MNIST$\rightarrow$USPS case. All the metrics are much worse than those achieved by our general OT method with the class-guided functional $\mathcal{F}_G$, see Table 1, 2 for comparison.

[8] github.com/iamalexkorotin/Wasserstein2Benchmark

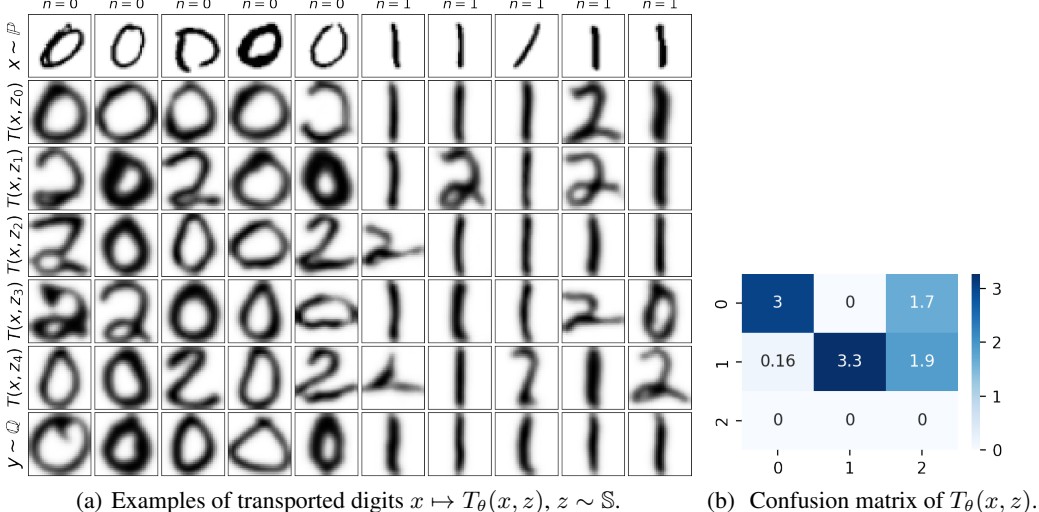

(a) Examples of transported digits $x \mapsto T_\theta(x, z)$, $z \sim \mathbb{S}$.  (b) Confusion matrix of $T_\theta(x, z)$.

Figure 15: Imbalanced MNIST $\rightarrow$ USPS translation with functional $\mathcal{F}_G$ (stochastic, with $z$).

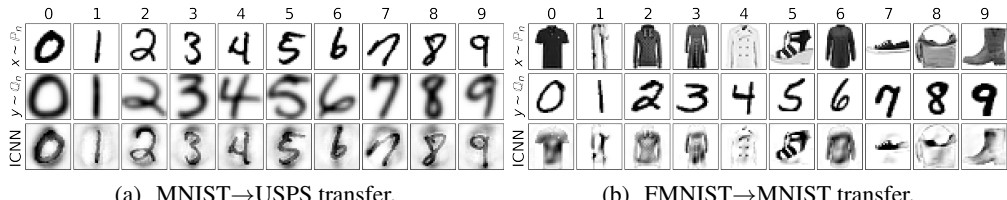

(a) MNIST→USPS transfer.        (b) FMNIST→MNIST transfer.

Figure 16: Results of ICNN-based method applied to the dataset transfer task.

### C.10 CLASSIC COST OT FOR DATASET TRANSFER

Our general cost functional-based algorithm can use **both labeled** and **unlabeled** target samples for training, which can be useful for the data transfer tasks. Existing continuous OT approaches do not handle a such type of training. Indeed, suppose we have additional information (labels) in the dataset and try to solve the class-guided mapping using the ICNN-based (Amos et al., 2017) $\mathbb{W}_2$ algorithms (Korotin et al., 2023b; Fan et al., 2023). In this scenario, we can train OT using only the labeled samples (10 separate maps in case of MNIST). The **unlabeled data immediately becomes useless**. Indeed, using unlabeled data for a class during training for that class implies that we know the labels for that class, which is a contradiction.

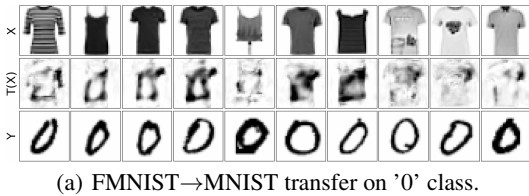

(a) FMNIST→MNIST transfer on '0' class.

Figure 17: Results of classic OT ($\mathbb{W}_2$) method applied to the dataset transfer task.

For illustrative purposes, we performed these experiments using the $\mathbb{W}_2$ algorithm for one of the clases on the FMNIST to MNITS mapping problem. We clearly see that the qualitative results are not competitive with our algorithm. This is because $\mathbb{W}_2$ is forced to train with only 10 target samples, the only labeled target samples in the problem setup.

### C.11 NON-DEFAULT CLASS CORRESPONDENCE

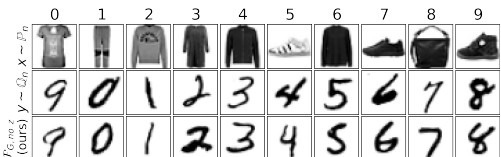

Figure 18: FMNIST→MNIST mapping with $\mathcal{F}_G$ no $z$ cost, classes are permuted.

To show that our method can work with any arbitrary correspondence between datasets, we also consider FMNIST→MNIST dataset transfer with the following non-default correspondence between the dataset classes:

$$0 \to 9, 1 \to 0, 2 \to 1, 3 \to 2, 4 \to 3, 5 \to 4, 6 \to 5, 7 \to 6, 8 \to 7, 9 \to 8.$$

In this experiment, we use the same architectures and data preprocessing as in dataset transfer tasks; see Appendix C.1. We use our $\mathcal{F}_G$ (7) as the cost functional and learn a deterministic transport map $T$ (no $z$). In this setting, our method produces comparable results to the previously reported in Section 6 accuracy equal to **83.1**, and FID **6.69**. The qualitative results are given in Figure 18.

### C.12 IN DOMAIN CLASS-PRESERVING

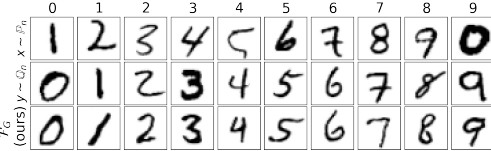

Figure 19: MNIST→MNIST mapping with $\mathcal{F}_G$, classes are permuted.

To provide an additional illustration, we also consider MNIST→MNIST dataset transfer with the following non-default correspondence between the dataset classes:

$$0 \to 9, 1 \to 0, 2 \to 1, 3 \to 2, 4 \to 3, 5 \to 4, 6 \to 5, 7 \to 6, 8 \to 7, 9 \to 8.$$

In this experiment, we use the same architectures, data preprocessing and metrics as in dataset transfer tasks (C.1). We use our $\mathcal{F}_G$ (7) as the cost functional and learn a transport map $T$. The resulted accuracy is equal to **95.1**, and FID **3.35**. The qualitative results are given in Figure 19.

### C.13 SOLVING BATCH EFFECT

The batch effect is a well-known issue in biology, particularly in high-throughput genomic studies such as gene expression microarrays, RNA-seq, and proteomics (Leek et al., 2010). It occurs when non-biological factors, such as different processing times or laboratory conditions, introduce systematic variations in the data. Addressing batch effects is crucial for ensuring robust and reproducible findings in biological research (Lazar et al., 2013). By solving this problem using our method, directly in the input space, we can preserve the samples' intrinsic structure, minimizing artifacts, and ensuring *biological* validation.

In our experiments, we map classes across two domains: *TM-baron-mouse-for-segerstolpe* and *segerstolpe-human*, consisting of 3,329 and 2,108 samples, respectively. The data was generated by the Splatter package (Zappia et al., 2017). Each domain consists of eight classes. The source domain $\mathbb{P}$ is fully-labelled, and the target $\mathbb{Q}$ contains 10 labelled samples per class. Each sample is a 657-sized vector pre-processed with default Splatter settings. (Zappia et al., 2017).

We employed feed-forward networks with one hidden layer of size 512 for the map $T_\theta$ and a hidden layer of size 1024 for the potential network $v_\omega$. To evaluate the accuracy, we trained single-layer neural network classifiers with soft-max output activation, using the available target data. Our method improved accuracy from **63.0** $\to$ **92.5**. Meanwhile, the best DOT solver (EMD) identified through search, as described in Appendix C.1, reduced accuracy from **63.0** $\to$ **50.4**.

# D  GENERAL FUNCTIONALS WITH CONDITIONAL INTERACTION ENERGY REGULARIZER

Generally speaking, for practically useful general cost functionals $\mathcal{F} : \mathcal{M}(\mathcal{X} \times \mathcal{Y}) \to \mathbb{R} \cup \{+\infty\}$ it may be difficult or even impossible to establish their strict or strong convexity. For instance, our considered class-guided functional $\mathcal{F}_G$ (7) is not necessarily strictly convex. In such cases, the maps $T^*$ which solve (5) are not necessarily stochastic OT maps, and our duality gap analysis (Theorem 3) is not directly applicable.

In this section, we propose a generic way to overcome this problem by means of strongly convex regularizers. Let $\mathcal{F}, \mathcal{R} : \mathcal{M}(\mathcal{X} \times \mathcal{Y}) \to \mathbb{R} \cup \{+\infty\}$ be convex, lower semi-continuous functionals, which are equal to $+\infty$ on $\mu \in \mathcal{M}(\mathcal{X} \times \mathcal{Y}) \setminus \mathcal{P}(\mathcal{X} \times \mathcal{Y})$. Additionally, we will assume that $\mathcal{R}$ is $\beta$-*strongly* convex on $\Pi(\mathbb{P})$ in some metric $\rho(\cdot, \cdot)$. For $\gamma > 0$, one may consider the following $\mathcal{R}$-regularized general OT problem:

$$\inf_{\pi \in \Pi(\mathbb{P}, \mathbb{Q})} \left\{ \mathcal{F}(\pi) + \gamma \mathcal{R}(\pi) \right\}.$$

Note that $\pi \mapsto \mathcal{F}(\pi) + \gamma \mathcal{R}(\pi)$ is convex, lower semi-continuous, separately *-increasing (since it equals $+\infty$ outside $\pi \in \mathcal{P}(\mathcal{X} \times \mathcal{Y})$, see the proof of Theorem 4 in Appendix A) and $\beta\gamma$-*strongly* convex on $\Pi(\mathbb{P})$ in $\rho(\cdot, \cdot)$. In the considered setup, functional $\mathcal{F}$ corresponds to a real problem a practitioner may want to solve, and functional $\mathcal{R}$ is the regularizer which slightly shifts the resulting solution but induces nice theoretical properties. Our proposed technique resembles the idea of the Neural Optimal Transport with Kernel Variance (Korotin et al., 2023a). In this section, we generalize their approach and make it applicable to our duality gap analysis (Theorem 3). Below we introduce an example of a strongly convex regularizer. Corresponding practical demonstrations are left to Appendix D.1.

**Conditional interaction energy functional.** Let $(\mathcal{Y}, l)$ be a semimetric space of negative type (Sejdinovic et al., 2013, §2.1), i.e. $l : \mathcal{Y} \times \mathcal{Y} \to \mathbb{R}$ is the semimetric and $\forall N \geq 2$, $y_1, y_2, \ldots, y_N \in \mathcal{Y}$ and $\forall \alpha_1, , \alpha_2, \ldots \alpha_N \in \mathbb{R}$ such that $\sum_{n=1}^{N} \alpha_n = 0$ it holds $\sum_{n=1}^{N} \sum_{n'=1}^{N} \alpha_n \alpha_{n'} l(y_n, y_{n'}) \leq 0$. The (square of) energy distance $\mathcal{E}_l$ w.r.t. semimetric $l$ between probability distributions $\mathbb{Q}_1, \mathbb{Q}_2 \in \mathcal{P}(\mathcal{Y})$ is ((Sejdinovic et al., 2013, §2.2)):

$$\mathcal{E}_l^2(\mathbb{Q}_1, \mathbb{Q}_2) = 2\mathbb{E}l(Y_1, Y_2) - \mathbb{E}l(Y_1, Y_1') - \mathbb{E}l(Y_2, Y_2'), \tag{38}$$

where $Y_1, Y_1' \sim \mathbb{Q}_1$; $Y_2, Y_2' \sim \mathbb{Q}_2$. Note that for $l(y, y') = \frac{1}{2}\|y - y'\|_2$ formula (38) reduces to (8). The energy distance is known to be a metric on $\mathcal{P}(\mathcal{Y})$ (Klebanov et al., 2005) (note that $\mathcal{Y}$ is compact). The examples of semimetrics of negative type include $l(y, y') = \|x - y\|_p^{\min\{1, p\}}$ for $0 < p \leq 2$ (Meckes, 2013, Th. 3.6).

Consider the following generalization of energy distance $\mathcal{E}_l$ on space $\Pi(\mathbb{P})$. Let $\pi_1, \pi_2 \in \Pi(\mathbb{P})$.

$$\rho_l^2(\pi_1, \pi_2) \stackrel{\text{def}}{=} \int_{\mathcal{X}} \mathcal{E}_l^2(\pi_1(\cdot|x), \pi_2(\cdot|x)) d\mathbb{P}(x). \tag{39}$$

**Proposition 2.** *It holds that $\rho_l(\cdot, \cdot)$ is a metric on $\Pi(\mathbb{P})$.*

*Proof of Proposition 2.* Obviously, $\forall \pi \in \Pi(\mathbb{P}) : \rho_l(\pi, \pi) = 0$ and $\forall \pi_1, \pi_2 \in \Pi(\mathbb{P}) : \rho_l(\pi_1, \pi_2) = \rho_l(\pi_2, \pi_1) \geq 0$. We are left to check the triangle inequality. Consider $\pi_1, \pi_2, \pi_3 \in \Pi(\mathbb{P})$. In what follows, for $\pi \in \Pi(\mathbb{P})$ and $x \in \mathcal{X}$, we denote the conditional distribution $\pi(\cdot|x)$ as $\pi^x$:

$$\rho_l(\pi_1, \pi_2) + \rho_l(\pi_2, \pi_3) \geq \rho_l(\pi_1, \pi_3) \Leftrightarrow$$
$$(\rho_l(\pi_1, \pi_2) + \rho_l(\pi_2, \pi_3))^2 \geq \rho_l^2(\pi_1, \pi_3) \Leftrightarrow$$

$$\int_{\mathcal{X}} \mathcal{E}_l^2(\pi_1^x, \pi_2^x) d\mathbb{P}(x) + \int_{\mathcal{X}} \mathcal{E}_l^2(\pi_2^x, \pi_3^x) d\mathbb{P}(x) + 2\sqrt{\int_{\mathcal{X}} \mathcal{E}_l^2(\pi_1^x, \pi_2^x) d\mathbb{P}(x) \int_{\mathcal{X}} \mathcal{E}_l^2(\pi_2^x, \pi_3^x) d\mathbb{P}(x)} \geq \int_{\mathcal{X}} \mathcal{E}_l^2(\pi_1^x, \pi_3^x) d\mathbb{P}(x) \Leftarrow \tag{40}$$

$$\int_{\mathcal{X}} \mathcal{E}_l^2(\pi_1^x, \pi_2^x) d\mathbb{P}(x) + \int_{\mathcal{X}} \mathcal{E}_l^2(\pi_2^x, \pi_3^x) d\mathbb{P}(x) + 2\int_{\mathcal{X}} \mathcal{E}_l(\pi_1^x, \pi_2^x)\mathcal{E}_l(\pi_2^x, \pi_3^x) d\mathbb{P}(x) \geq \int_{\mathcal{X}} \mathcal{E}_l^2(\pi_1^x, \pi_3^x) d\mathbb{P}(x) \Leftrightarrow$$

$$\int_{\mathcal{X}} \underbrace{\left[ (\mathcal{E}_l(\pi_1^x, \pi_2^x) + \mathcal{E}_l(\pi_2^x, \pi_3^x))^2 - \mathcal{E}_l^2(\pi_1^x, \pi_3^x) \right]}_{\geq 0 \text{ due to triangle inequality for } \mathcal{E}_l} d\mathbb{P}(x) \geq 0,$$

where in line (40) we apply the Cauchy–Bunyakovsky inequality (Bouniakowsky, 1859):

$$\int_{\mathcal{X}} \mathcal{E}_l^2(\pi_1^x, \pi_2^x) d\mathbb{P}(x) \int_{\mathcal{X}} \mathcal{E}_l^2(\pi_2^x, \pi_3^x) d\mathbb{P}(x) \geq \left( \int_{\mathcal{X}} \mathcal{E}_l(\pi_1^x, \pi_2^x) \mathcal{E}_l(\pi_2^x, \pi_3^x) d\mathbb{P}(x) \right)^2.$$

$\square$

Now we are ready to introduce our proposed strongly convex (w.r.t. $\rho_l$) regularizer. Let $\pi \in \Pi(\mathcal{X})$ and $l$ be a semimetric on $\mathcal{Y}$ of negative type. We define

$$\mathcal{R}_l(\pi) = -\frac{1}{2} \int_{\mathcal{X}} \int_{\mathcal{Y}} \int_{\mathcal{Y}} l(y, y') d\pi^x(y) d\pi^x(y') d\mathbb{P}(x). \tag{41}$$

We call $\mathcal{R}_l$ to be *conditional interaction energy functional*. In the context of solving the OT problem, it was first introduced in (Korotin et al., 2023a) from the perspectives of RKHS and kernel embeddings (Sejdinovic et al., 2013, §3). The authors of (Korotin et al., 2023a) establish the conditions under which the semi-dual (max-min) formulation of weak OT problem regularized with $\mathcal{R}_l$ yields the unique solution, i.e., they deal with the *strict* convexity of $\mathcal{R}_l$. In contrast, our paper exploits *strong* convexity and provides the additional error analysis (Theorem 3) which helps with tailoring theoretical guarantees to actual practical procedures for arbitrary strongly convex functionals. Below, we prove that $\mathcal{R}_l$ is strongly convex w.r.t. $\rho_l$ and, under additional assumptions on $l$, is lower semi-continuous.

**Proposition 3.** $\mathcal{R}_l$ *is 1-strongly convex on* $\Pi(\mathbb{P})$ *w.r.t.* $\rho_l$.

*Proof of Proposition 3.* Let $\pi_1, \pi_2 \in \Pi(\mathbb{P}), 0 \leq \alpha \leq 1$. Consider the left-hand side of (21):

$$\mathcal{R}_l(\alpha \pi_1 + (1 - \alpha)\pi_2) =$$

$$-\frac{1}{2} \int_{\mathcal{X}} \int_{\mathcal{Y} \times \mathcal{Y}} l(y, y') d[\alpha \pi_1 + (1 - \alpha)\pi_2]^x(y) d[\alpha \pi_1 + (1 - \alpha)\pi_2]^x(y') d\mathbb{P}(x) =$$

$$-\frac{1}{2} \alpha^2 \int_{\mathcal{X}} \int_{\mathcal{Y} \times \mathcal{Y}} l(y, y') d\pi_1^x(y) d\pi_1^x(y') d\mathbb{P}(x) +$$

$$-\alpha(1 - \alpha) \int_{\mathcal{X}} \int_{\mathcal{Y} \times \mathcal{Y}} l(y, y') d\pi_1^x(y) d\pi_2^x(y') d\mathbb{P}(x) +$$

$$-\frac{1}{2} (1 - \alpha)^2 \int_{\mathcal{X}} \int_{\mathcal{Y} \times \mathcal{Y}} l(y, y') d\pi_2^x(y) d\pi_2^x(y') d\mathbb{P}(x) =$$

$$\left( -\frac{1}{2}\alpha + \frac{1}{2}\alpha(1 - \alpha) \right) \int_{\mathcal{X}} \int_{\mathcal{Y} \times \mathcal{Y}} l(y, y') d\pi_1^x(y) d\pi_1^x(y') d\mathbb{P}(x) +$$

$$-\alpha(1 - \alpha) \int_{\mathcal{X}} \int_{\mathcal{Y} \times \mathcal{Y}} l(y, y') d\pi_1^x(y) d\pi_2^x(y') d\mathbb{P}(x) +$$

$$\left( -\frac{1}{2}(1 - \alpha) + \frac{1}{2}\alpha(1 - \alpha) \right) \int_{\mathcal{X}} \int_{\mathcal{Y} \times \mathcal{Y}} l(y, y') d\pi_2^x(y) d\pi_2^x(y') d\mathbb{P}(x) =$$

$$\underbrace{-\frac{1}{2}\alpha \int_{\mathcal{X}} \int_{\mathcal{Y} \times \mathcal{Y}} l(y, y') d\pi_1^x(y) d\pi_1^x(y') d\mathbb{P}(x)}_{=\alpha \mathcal{R}_l(\pi_1)} \underbrace{-\frac{1}{2}(1 - \alpha) \int_{\mathcal{X}} \int_{\mathcal{Y} \times \mathcal{Y}} l(y, y') d\pi_2^x(y) d\pi_2^x(y') d\mathbb{P}(x)}_{=(1-\alpha)\mathcal{R}_l(\pi_2)} +$$

$$\frac{\alpha(1-\alpha)}{2} \int_{\mathcal{X}} \underbrace{\left( \int_{\mathcal{Y} \times \mathcal{Y}} l(y, y') d\pi_1^x(y) d\pi_1^x(y') + \int_{\mathcal{Y} \times \mathcal{Y}} l(y, y') d\pi_2^x(y) d\pi_2^x(y') - 2 \int_{\mathcal{Y} \times \mathcal{Y}} l(y, y') d\pi_1^x(y) d\pi_2^x(y') \right)}_{=-\mathcal{E}_l(\pi_1^x, \pi_2^x)} d\mathbb{P}(x) =$$

$$\alpha \mathcal{R}_l(\pi_1) + (1 - \alpha)\mathcal{R}_l(\pi_2) - \frac{1}{2}\alpha(1 - \alpha)\rho_l^2(\pi_1, \pi_2),$$

i.e., $\mathcal{R}_l(\alpha \pi_1 + (1-\alpha)\pi_2) = \alpha \mathcal{R}_l(\pi_1) + (1-\alpha)\mathcal{R}_l(\pi_2) - \frac{\alpha(1-\alpha)}{2}\rho_l^2(\pi_1, \pi_2)$, which finishes the proof. $\square$

**Proposition 4.** *Assume that $l$ is continuous (it is the case for all reasonable semimetrics $l$). Then $\mathcal{R}_l$ is lower semi-continuous on $\Pi(\mathbb{P})$.*

*Proof of Proposition 4.* Consider the functional $\mathcal{W}_l : \mathcal{X} \times \mathcal{P}(\mathcal{Y}) \to \mathbb{R} \cup \{+\infty\}$:

$$\mathcal{W}_l(x, \mu) = -\int_{\mathcal{Y} \times \mathcal{Y}} l(y, y') d\mu(y) d\mu(y'),$$

then the conditional interaction energy functional could be expressed as follows: $\mathcal{R}_l(\pi) = \frac{1}{2} \int_{\mathcal{X}} \mathcal{W}_l(x, \pi^x) d\mathbb{P}(x)$. We are to check that $\mathcal{W}_l$ satisfies Condition (A+) in (Backhoff-Veraguas et al., 2019, Definition 2.7). Note that $\mathcal{W}_l$ actually does not depend on $x \in \mathcal{X}$.

- The *lower-semicontinuity* of $\mathcal{W}_l$ follows from (Santambrogio, 2015, Proposition 7.2) and the equivalence of the weak convergence and the convergence w.r.t. Wasserstein metric on $\mathcal{P}(\mathcal{Y})$ where $\mathcal{Y}$ is compact, see (Villani, 2008, Theorem 6.8).

- Since $(y, y') \mapsto -l(y, y')$ is a lower-semicontinuous, it achieves its minimum on the compact $\mathcal{Y} \times \mathcal{Y}$ which *lower-bounds* the functional $\mathcal{W}_l$.

- The convexity (even 1-strong convexity w.r.t. metric $\mathcal{E}_l(\cdot, \cdot)$) of functional $\mathcal{W}_l$ was de facto established in Proposition 3. In particular, given $\mu_1, \mu_2 \in \mathcal{P}(\mathcal{Y})$, $\alpha \in (0, 1)$, $x \in \mathcal{X}$ it holds:

$$\mathcal{W}_l(x, \alpha\mu_1 + (1-\alpha)\mu_2) = \alpha\mathcal{W}_l(x, \mu_1) + (1-\alpha)\mathcal{W}_l(x, \mu_2) - \frac{\alpha(1-\alpha)}{2}\mathcal{E}_l^2(\mu_1, \mu_2).$$

The application of (Backhoff-Veraguas et al., 2019, Proposition 2.8, Eq. (2.16)) finishes the proof. □

### D.1 EXPERIMENTS WITH CONDITIONAL INTERACTION ENERGY REGULARIZER

In the previous Section D, we introduce an example of the strongly convex regularizer. In this section, we present experiments to investigate the impact of strongly convex regularization on our general cost functional $\mathcal{F}_G$. In particular, we conduct experiments on the FMNIST-MNIST dataset transfer problem using the proposed conditional interaction energy regularizer with $l(y, y') = \|y - y'\|_2$. To empirically estimate the impact of the regularization, we test different coefficients $\gamma \in [0.001, 0.01, 0.1]$. The results are shown in the following Figure 20 and Table 6.

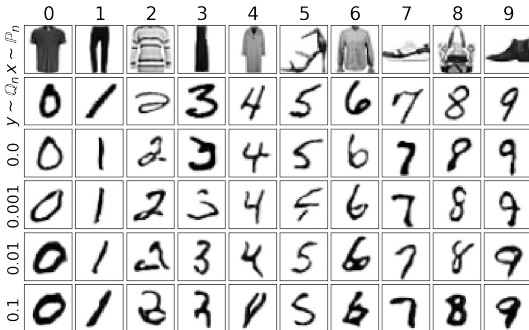

Figure 20: Qualitative results of the FMNIST→MNIST mapping with $\mathcal{F}_G$ cost and using different values $\gamma$ of the conditional interaction energy regularization.

| $\gamma$ | 0 | 0.001 | 0.01 | 0.1 |
|---|---|---|---|---|
| Accuracy↑ | 83.33 | 81.87 | 79.47 | 65.11 |
| FID ↓ | 5.27 | 7.67 | 3.95 | 7.33 |

Table 6: Accuracy↑ and FID ↓ of the map learned on FMNIST→MNIST with $\mathcal{F}_G$ cost and different values $\gamma$ of the conditional interaction energy regularization.

It can be seen that the *small* amount of regularization ($\gamma = 0.001$) does not affect the results. But *high* values decrease the accuracy, which is expected because the regularization contradicts the dataset transfer problem. Increasing the value of $\gamma$ shifts the solution to be more diverse instead of matching the classes.

# E    PAIR-GUIDED COST FUNCTIONAL

## E.1    ALGORITHM

Recall that in our main manuscript we parameterize the learned plan $\pi$ via stochastic map $[x, T(x, z)]$, $x \sim \mathbb{P}, z \sim \mathbb{S}$. In practice, we found that substitution $T(x, z)$ with a *deterministic* map $T(x)$ generally improves the results in the case of pair-guided cost functional. This is possibly due to the paired nature of the considered problem. Therefore, we adapt our proposed Algorithm 3 for deterministic map $T(x)$ and report all metrics and demonstrations exactly for this setup.

---

**Algorithm 3:** Neural optimal transport with pair-guided cost functional and deterministic map

---

**Input**    : Distributions $\mathbb{P}, \mathbb{Q}$ accessible by samples; paired data set $(x_1, y^*(x_1)), \dots$
$\dots, (x_N, y^*(x_N))$, where $x_{1:N} \sim \mathbb{P}$ and $y^*(x_{1:N}) \sim \mathbb{Q}$; mapping network
$T_\theta : \mathbb{R}^P \to \mathbb{R}^Q$; potential network $v_\omega : \mathbb{R}^Q \to \mathbb{R}$; number of inner iterations $K_T$;

**Output** : Learned OT map $T_\theta$ representing (pair-guided) OT plan between distributions $\mathbb{P}, \mathbb{Q}$;

**repeat**

  Sample batches $Y \sim \mathbb{Q}, X \sim \mathbb{P}$;

  $\mathcal{L}_v \leftarrow \sum_{x \in X} \frac{v_\omega(T_\theta(x))}{|X|} - \sum_{y \in Y} \frac{v_\omega(y)}{|Y|}$;

  Update $\omega$ by using $\frac{\partial \mathcal{L}_v}{\partial \omega}$;

  **for** $k_T = 1, 2, \dots, K_T$ **do**

    Sample batch $X_d$ from the paired data set;

    $\mathcal{L}_T \leftarrow \sum_{x_d \in X_d} \frac{\ell(T_\theta(x_d), y^*(x_d))}{|X_d|} - \sum_{x_d \in X_P} \frac{v_\omega(T_\theta(x_d))}{|X_d|}$;

    Update $\theta$ by using $\frac{\partial \mathcal{L}_T}{\partial \theta}$;

**until** not converged;

---

## E.2    DATASETS

**Comic-Faces-V1**[9]**:** This dataset contains paired samples which are useful for real-to-comic convertion. The original resolution is 512x512, 10000 pairs (total 20k images)

**Edges-to-Shoes:** This dataset consists of 50,025 shoe images and their corresponding edges split into train and test subsets.

**CelebAMask-HQ**[10]**:** This is a large-scale face image dataset that has 30,000 high-resolution face images selected from CelebA-HQ dataset. Each image has segmentation mask of facial attributes corresponding to CelebA. The masks of CelebAMask-HQ were manually-annotated with the size of 512 x 512 images.

## E.3    TRAINING DETAILS

In our experiments, we compare several methods. As the baselines we consider (unsupervised) NOT, and RMSE regression. Here use U2Net [11] as the transport map $T_\theta(x)$ (generator). For comparison with NOT, we use their publicly available code [12]. There we employ the (unsupervised) RMSE as the cost function (the method is denoted by $\mathbb{W}_1$ in the table and figures). Note that unsupervised NOT (Figures 4 and 21) fails to perform the translation in both the cases. For the comparison with Pix2Pix, we use the official implementations with the default hyperparameters [13].

To train our model, we use Adam (Kingma & Ba, 2014) optimizer with $lr = 10^{-4}$ for both $T_\theta$ and $v_\omega$. The number of inner iterations for $T_\theta$ is $K_T = 10$. The batch size of $K_B = 8$ was set for the comic-faces and shoes experiments. The batch size of $K_B = 32$ was used for CelebAMask-HQ. Our method converges in $\approx$ 60k iterations of $v_\omega$.

---

[9] https://www.kaggle.com/datasets/defileroff/comic-faces-paired-synthetic

[10] https://github.com/switchablenorms/CelebAMask-HQ

[11] https://github.com/xuebinqin/U-2-Net

[12] https://github.com/iamalexkorotin/NeuralOptimalTransport

[13] https://github.com/junyanz/pytorch-CycleGAN-and-pix2pix

| **Datasets** $(256 \times 256)$ | $\mathbb{W}_1$ | **RMSE** | **Pix2Pix** | $\mathcal{F}_s$ **[Ours]** |
|---|---|---|---|---|
| Comic-Faces-V1 | $> 100$ | 79.16 | 37.02 | **35.42** |
| Edges-to-Shoes | $> 100$ | 61.55 | - | **49.53** |

Table 7: FID $\downarrow$ of the maps learned by the translation methods in view.

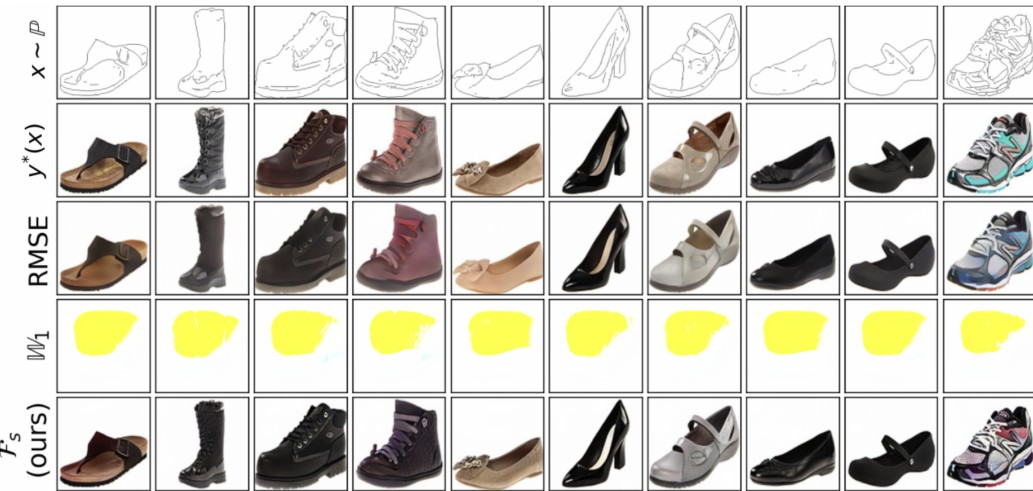

Figure 21: Results for Edges-to-Shoes with the Pair-guided cost, images resolution is $256 \times 256$.

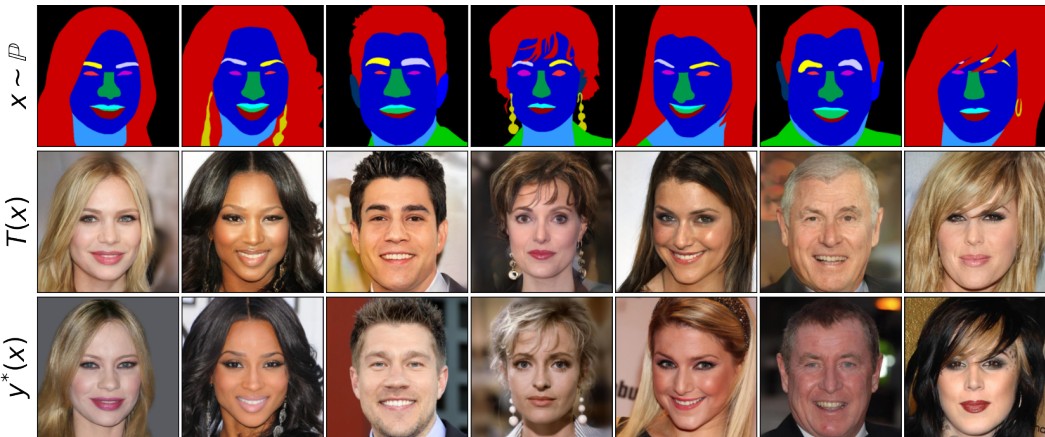

Figure 22: Results for CelebAMask with the VGG-based perceptual Pair-guided cost, images resolution is $256 \times 256$.

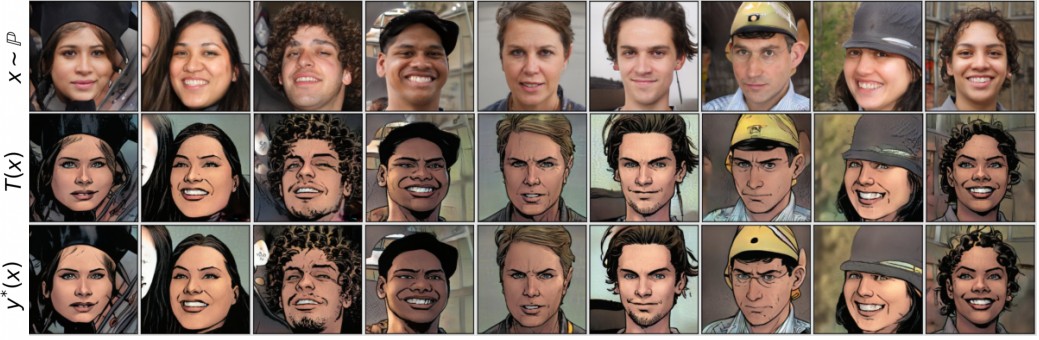

Figure 23: Results for Comic-Faces-V1 with the Pair-guided cost, images resolution is $512 \times 512$.

