# OpenReview forum: "Neural Optimal Transport with General Cost Functionals"
_ICLR.cc/2024/Conference — ICLR 2024 poster_

### Official Review · Reviewer_BH6f · 2023-10-29

**Soundness:** 3 good
**Presentation:** 3 good
**Contribution:** 3 good
**Rating:** 6
**Confidence:** 3

**Summary:**

This paper proposes a general OT formulation, which uses a functional \mathcal{F} to encompass common objectives and regularizers as special cases. The authors introduced a method for addressing the continuous general OT problem and illustrated how a general functional \mathcal{F} can incorporate information, such as the presence of class labels in the data. To validate the method, the authors use synthetic datasets and various MNIST datasets for testing.

**Strengths:**

The closest prior work appears to be the study by Korotin et al. (2023b), as cited in the paper. Building upon this previous research, the authors demonstrate how one can preserve the class-label structure in OT. This contribution is novel to my knowledge.

**Weaknesses:**

As I understand it, one benefit of employing a general functional F is to account for the class-label structure. Are there any other intended applications of a general F? If the sole purpose is to consider the class-label structure, perhaps some proofs (e.g., the proof to Theorem 3) could be simplified.

I do not fully understand the image data experiments depicted in Fig 3(a) and (b) in section 5.2, and I would appreciate it if the authors could provide further explanation. From my understanding, the goal here is to identify an optimal transport (OT) map between two data distributions (e.g., the distribution of MNIST and KMNIST images) while preserving the class correspondence. To achieve this, one could visualize several source images from the same class in the source dataset and check if the corresponding target images are from the same class. However, Fig 3 appears to display only a single source image per class (top rows) along with a target image per class (2nd rows), making it unclear whether the class correspondence is preserved overall.

Another reason why Figures 3(a) and 3(b) are challenging to understand is the absence of a natural correspondence between the classes of MNIST -> KMNIST and FMNIST -> MNIST images. It is thus difficult to check the correspondence visually. To enhance visualization, would it be reasonable to use only a single dataset, such as MNIST, and establish the correspondence of images from class 0, 1, 2, ..., 9 to a permuted class order, such as 1, 2, 3, ..., 9, 0? This approach would help to visualize the class correspondence.

**Questions:**

See the "Weaknesses" section above.

---

> ### Author Response · Authors · 2023-11-20
> **Thanks for your review**
>
> Thank you for your comments and questions. Please find the answers to your questions below.
>
> **Q1: (...) Are there any other intended applications of a general F? (...)**
>
> The primary advantage of the general cost is its ability to incorporate additional information beyond just class labels. We added an additional example of a general cost functional, please refer to the **general answer for all the reviewers** and **Appendix E**.
>
> **Q2: I do not fully understand the image data experiments depicted in Fig 3(a) and (b) in section 5.2, and I would appreciate it if the authors could provide further explanation.  (...) one could visualize several source images from the same class in the source dataset and check if the corresponding target images are from the same class. (...)**
>
> Yes, your understanding is correct. Please note that the key goal of this visualization is to show that the proposed method generates images without noise and artefacts. The fact that the class if preserved is assessed via the **quantitative  metric** (*accuracy*), see Table 1.2. Following your comment, we have provided **an additional illustration** for 3 input images per class on the MNIST $\to$ FMNIST dataset. These results are included in **Appendix C.4**. This additional visualization aims to address the concern and provide a clearer qualitative examples of how our method preserves class correspondence across multiple input images per class.
>
> **Q3: (...) To enhance visualization, would it be reasonable to use only a single dataset, such as MNIST, and establish the correspondence of images from class 0, 1, 2, ..., 9 to a permuted class order, such as 1, 2, 3, ..., 9, 0? This approach would help to visualize the class correspondence.**
>
> We appreciate your suggestion on presentation improvements. We conducted experiments using a single dataset, specifically MNIST. We established the shifted correspondence of images from classes $0\to9, 1\to0, 2\to1, 3\to2, 4\to3, 5\to4, 6\to5, 7\to6, 8\to7, 9\to8$. The results of these experiments have been included in **Appendix C.11** to enhance visualization. It's worth mentioning that, in addition to mapping unrelated domains, we also explored related domain experiments, as detailed in **Appendix C.3**.
>
> **Concluding remarks**: Please respond to our post to let us know if the clarifications above suitably address your concerns about our work. We are happy to address any remaining points during the discussion phase; if the responses above are sufficient, we kindly ask that you consider raising your score.

---

### Official Review · Reviewer_iaG1 · 2023-10-30

**Soundness:** 3 good
**Presentation:** 2 fair
**Contribution:** 3 good
**Rating:** 6
**Confidence:** 3

**Summary:**

This paper introduces a novel neural network-based algorithm for computing optimal transport plans (OT) for cost functionals that go beyond typical Euclidean costs like $\ell^1$ or $\ell^2$. These functionals offer greater flexibility and allow the incorporation of auxiliary information, such as class labels, in constructing the transport map.

Existing methods for general costs are discrete and lack out-of-sample estimation capabilities. The paper addresses the challenge of designing a continuous OT approach for general costs that can generalize to new data points in high-dimensional spaces like images. Additionally, it provides theoretical error analysis for the recovered transport plans.

As an application, the paper demonstrates how to construct a cost functional that maps data distributions while preserving class-wise structures.

**Strengths:**

- The paper is well-structured with a clear motivation, thorough literature review, rigorous theoretical analysis, comprehensive numerical experiments, detailed implementation, and insightful discussions.
- The paper expands upon existing neural OT techniques to accommodate general cost functionals, offering potential applications in mapping data distributions while maintaining class-wise structures.

**Weaknesses:**

- The paper heavily draws upon the prior work of (Korotin et al. 2023a) for its theoretical foundations. Approximately 5 out of 9 pages are dedicated to presenting these theoretical results, which may not constitute the primary novelty of the paper. Based on my personal reading, it seems that the authors might be overselling their theoretical contributions, potentially leading to a less reader-friendly introduction.
- The paper exceeds the strict upper limit of 9 pages for the main text of the submission by including a section on reproducibility (Section 7) on Page 10. As a reviewer, this doesn't pose an issue for me, but I would advise adhering to the prescribed page limits as a matter of following the rules.


[Korotin et al. 2023a] Alexander Korotin, Daniil Selikhanovych, and Evgeny Burnaev. Kernel neural optimal transport. In
International Conference on Learning Representations, 2023a.

**Questions:**

- A direct comparison of the theoretical findings with those presented in (Korotin et al. 2023a) is essential. Readers are likely seeking a consolidated presentation rather than having to review two separate papers with overlapping theoretical content.
- From my perspective, the paper's novelty appears to lie more in its practical applications, particularly in utilizing a cost functional to map data distributions while maintaining class-wise structures. In light of this, it would be beneficial for the paper to allocate more of its main text to discussing practical implementation aspects.
- From my own interests, I would find it valuable if the paper could delve further into potential applications involving general cost functionals.

---

> ### Author Response · Authors · 2023-11-20
> **Thanks for your review**
>
> Thank you for spending time reviewing our paper and providing valuable feedback that will help us improve the manuscript. Please find below the answers to your questions.
>
> **Q1: The paper heavily draws upon the prior work of (Korotin et al. 2023a) for its theoretical foundations. (...). Based on my personal reading, it seems that the authors might be overselling their theoretical contributions...**
>
> We appreciate the reviewer's observation and concerns, but we believe that the theorems, corollaries, and results presented in Sections 3 are novel and important. These results are form the theoretical backbone of our work and crucial in establishing a theoretically justified algorithm that surpasses prior OT methods in practice (Section 5) as it allows using additional label information. Moreover, the unique error analysis via duality gaps for the general cost in Section 4 represents a novel aspect of our approach, distinguishing it from existing methods.
>
> **Q2:The paper exceeds the strict upper limit of 9 pages for the main text of the submission by including a section on reproducibility (Section 7) on Page 10. (...).**
>
> Please note that according to ICLR 2024 rules, it is allowed to include the reproducibility statement as an additional 10th page, see the reproducibility section in https://iclr.cc/Conferences/2024/AuthorGuide.
>
> **Q3:A direct comparison of the theoretical findings with those presented in (Korotin et al. 2023a) is essential. Readers are likely seeking a consolidated presentation rather than having to review two separate papers with overlapping theoretical content.**
>
> To be honest, we do not clearly understand your concern. Korotin et. al. (2023a) proposes kernel regularizers which we use as an example of a strongly convex regularizer in **Appendix D** of our paper. Korotin et. al. (2023a) uses only the *strict* convexity of this functional, while we derive its **strong** convexity for our purposes. Please see the detailed discussion before Proposition 3 (Page 28).
>
>
> **Q4: (...) the paper's novelty appears to lie more in its practical applications (...) it would be beneficial for the paper to allocate more of its main text to discussing practical implementation aspects.**
>
> Thank you for acknowledging our practical contribution. However, it is important to note, as mentioned in our response to Q1, that our theoretical contribution is equally significant. General costs constitute a fruitful field in discrete OT, and our work extends them to the continuous case, effectively bridging a gap between OT and deep learning. We have established theoretical properties that specifically arise in the continuous case. We believe that our analytical insights can prove valuable for future studies in neural optimal transport.
>
> The technical details are presented in **Appendix C.1** and the source code is in the supplementary material. *Could you please recommend, which practical implementation aspects to move to the main text from Appendix?*
>
>
> **Q5:From my own interests, I would find it valuable if the paper could delve further into potential applications involving general cost functionals.**
>
> Thank you for you suggestion. First, to demonstrate the flexibility of our proposed approach, we conducted additional experiments with the newly-introduced pair-guided cost functional, please refer to the **general answer for all the reviewers** and **Appendix E**.
>
> Second, as an example of a potential practical application, in the initial submission, we provided experiments on the batch effect problem. The batch effect is a well-known problem in biology, especially in high-throughput genomic studies such as gene expression microarrays, RNA-seq, and proteomics. See **Appendix I** for details.
>
> More generally, our training method holds promise in addressing a broad spectrum of problems, including transfer between different modalities such as audio-to-image [1,2] and fMRI-to-image [3]. Acquiring a fully labeled dataset for these types of problems is often costly, making it crucial to develop a method that utilizes labels to construct the required transformation. Our method can be directly applied to such challenges, and we believe that delving into these types of problems is an interesting direction for future research.
>
> **References:**
>
> (Korotin et al. 2023a). Kernel neural optimal transport. (In ICLR, 2023a)
>
> [1] CH Wan et al. Towards audio to scene image synthesis using generative adversarial network.
>
> [2] Chunjin Song et al. AudioViewer: Learning To Visualize Sounds.
>
> [3] Zijiao Chen et al. Seeing Beyond the Brain: Conditional Diffusion Model with Sparse Masked Modeling for Vision Decoding
>
> **Concluding remarks:** Please respond to our post to let us know if the clarifications above suitably address your concerns about our work. We are happy to address any remaining points during the discussion phase; if the responses above are sufficient, we kindly ask that you consider raising your score.

---

### Official Review · Reviewer_LT3k · 2023-10-30

**Soundness:** 3 good
**Presentation:** 3 good
**Contribution:** 3 good
**Rating:** 6
**Confidence:** 3

**Summary:**

The paper discusses the problem of computing optimal transportation plans for a general cost functional. In particular, the paper proposes a max-min formulation of the problem, provides theoretical consistency results, and a stochastic optimization algorithm to numerically solve it. The problem is motivated by a "dataset transfer problem" where class labels are required to be preserved with the transportation plan. The proposed algorithm is illustrated on an a toy example with moon dataset and an example involving image dataset.

**Strengths:**

- The topic of the paper is interesting and valuable to the researchers
- The paper is written with mathematical rigor
- The proposed algorithm is supported by theoretical arguments and numerical experiments
- The theoretical results are important and useful

**Weaknesses:**

1- The novelty of the paper, in comparison to Ref[1], is weak.
- The theoretical novelty, in comparison to the existing theoretical result in Ref[1] is not explained well (what is the new approach or new tool that is being used here). What are the challenges of considering a general cost functional that the previous approach could not handle.
- The computational algorithm is also very similar. The NOT algorithm block in Ref[1] can be simply extended to general cost functional. The proposed algorithm block is very similar, with the only difference that it is written for a class-guided functional.

2 - As I was reading the paper, I found the motivation, the theoretical discussion, and the numerical examples a bit disconnected. If the main contribution of the paper is the OT with general cost functional, it is necessary to provide several examples of a general cost functional, rather than just focusing on class guided cases. If the main goal of the paper is to do the class guided transportation, it should be reflected in the title, there should be more motivation why this is useful in practice, and what are the existing approaches for this particular problem.

3 - The comparison with the existing OT approaches does not seem fair as they do not optimize the same cost function as your apporach. A possible comparison is to use the existing OT algorithms to do to transportation for each class separately, resulting in 10 different maps (for the MNIST case) and discuss how your proposed method, which only trains one map, is computationally more efficient, while the loss in accuracy is not significant.

[1] Alexander Korotin, Daniil Selikhanovych, and Evgeny Burnaev. Neural optimal transport. In International Conference on Learning Representations, 2023

**Questions:**

- It is interesting to see and discuss how the algorithm performs when the data from classes overlap.
- Regarding the discussion at the beginning of Sec 3.2., why is there a "measurable" map for each coupling? I understand existence for each x, but not sure how to argue existence of a measurable map as a function of x and z.

---

> ### Author Response · Authors · 2023-11-20
> **Thanks for your review (Part 1)**
>
> Thank you for spending time reviewing our paper and providing useful feedback that will help us improve the manuscript. Please find the answers to your questions below.
>
> **Q1.1:  The novelty of the paper, in comparison to Ref[1] (...) what is the new approach or new theoretical tool that is being used here.**
>
> - When deriving our max-min duality formula for general OT (section 3.1/3.2), we do not introduce new principal tools which have not been considered before. We just derive a max-min reformulation of general OT; it generalizes previously known classical and weak costs Ref[1], see the discussion at the end of Section 3.2. This form enables us to establish an algorithm for general OT which **can be applied to problems where the predecessors are not directly applicable** (*see the answers to Q1.2, Q2 below*).
>
>
> - The **newly introduced theoretical tool** is the duality gap analysis of Section 3.3. Please see the discussion around the Theorem 3. Previously known error analysis works exclusively with the classical OT and operate only under certain restrictive assumptions such as the convexity of the dual potential. In contrast, our error analysis is free from assumptions on the dual variable and *is applicable* not only to general OT but also *to weak OT Ref[1], for which there is currently no existing error analysis*. Please reconsider the end of **Section 3.3** (relation to prior works) and **Appendix D**.
>
>
> **Q1.2. What are the challenges of considering a general cost functional that the previous approach could not handle?**
>
> Our general cost functional-based algorithm can use **both labeled** and **unlabeled** target samples for training, which can be useful for the data transfer tasks. Existing continuous OT approaches do not handle a such type of training. Indeed, suppose we have additional information (labels) in the dataset and try to solve the class-guided mapping using the Ref[1] (as you mentioned in your Q4 below). In this scenario, we can train Ref[1] using only the labeled samples (10 separate maps in case of MNIST). In this case, the **unlabeled data immediately becomes useless**. Indeed, using unlabeled data for a class during training for that class implies that we know the labels for that class, which is a contradiction. Thus, training Ref[1] using both labeled and unlabeled target samples, as in our approach, is not possible.
>
> **Q2. The computational algorithm is also very similar. The NOT algorithm block in Ref[1] can be simply extended to general cost functional. (...) the only difference that it is written for a class-guided functional.**
>
> We agree that our algorithms may look similar, but this is because Ref[1] is a special case of our general approach. Our algorithm provides a better flexibility for using continuous OT. As we write in **Section 4**, the **differences** lie is the estimation of the cost functional $\mathcal{F}$ to perform the gradient updates. This difference is crucial, because such estimation, for example, allows to exploit both labeled and unlabeled data (see the answer to your previous question).
>
> **Q3: (...) I found the motivation, the theoretical discussion, and the numerical examples a bit disconnected. (...) it is necessary to provide several examples of a general cost functional (...).**
>
> Our primary conceptual contribution involves extending continuous optimal transport to general cost functionals, not exclusively focusing on the class-guided costs.
> We appreciate your suggestion to include more examples of general cost functionals in the paper, and we have addressed this concern, please see **general response to all the reviewers**.
>
>
> **Q4: (...) A possible comparison is to use the existing OT algorithms to do to transportation for each class separately, resulting in 10 different maps (...).**
>
>
> In response to your suggestion, we conducted this experiments using the Ref[1] algorithm for single class separately. The results of these experiments are presented in **Appendix C.9**. We can see that the qualitative results are not competitive with our algorithm. This is because Ref[1] is forced to train using only 10 target samples (the only labeled target samples in the problem setup), see the **discussion in Q1.2, Q2 above**.

---

> > ### Author Response · Authors · 2023-11-20
> > **Part 2.**
> >
> > **Q5: It is interesting to see and discuss how the algorithm performs when the data from classes overlap.**
> >
> > To specifically address the performance in scenarios where classes overlap, we conducted a class-guided mapping task on a Gaussian dataset with overlapping classes. For more detailed insights, please refer to **Appendix C.2**. Our method effectively handles this scenario, showcasing the robustness of our model in dealing with overlapping classes.
> >
> > **Q6: Regarding the discussion at the beginning of Sec 3.2., why is there a "measurable" map for each coupling? I understand existence for each x, but not sure how to argue existence of a measurable map as a function of x and z.**
> >
> > This fact follows from [2]. In particular, Theorem 5.3 yields that the mapping $x \rightarrow \pi(\cdot \vert x)$ is a so-called probability kernel (see definition in [2]). After that, Lemma 2.22 establishes that there exist some measurable function $T : \mathcal{X}\times [0, 1] \rightarrow \mathcal{Y}$ (we utilize our notation here) such that $T(x, \cdot)\sharp U(0, 1)$ has distribution $\pi(\cdot \vert x)$. One can easily substitute the uniform distribution $U(0, 1)$ with our considered latent distribution $\mathbb{S}$.
> >
> > **Concluding remarks**: Please respond to our post to let us know if the clarifications above suitably address your concerns about our work. We are happy to address any remaining points during the discussion phase; if the responses above are sufficient, we kindly ask that you consider raising your score.
> >
> > **References:**
> >
> > [1] Alexander Korotin, Daniil Selikhanovych, and Evgeny Burnaev. Neural optimal transport. In International Conference on Learning Representations, 2023.
> >
> > [2] Kallenberg O., Foundations of Modern Probability, 1997 (https://doi.org/10.1007/b98838).

---

> > > ### Comment · Reviewer_LT3k · 2023-11-22
> > >
> > > I appreciate the author's response and answering my questions. I see the contribution of the paper as extending [1] to a general class of cost functions. The author's response did not show me any algorithmic or theoretical novelty.  I find the applications with the labeled data, and their additional experiments, very interesting. I wish the paper was written in a way that was more focused on the application, rather than claiming to propose a novel algorithm. I increase my score to above acceptance but I do not strongly support acceptance.

---

### Official Review · Reviewer_nKa4 · 2023-10-31

**Soundness:** 3 good
**Presentation:** 3 good
**Contribution:** 3 good
**Rating:** 6
**Confidence:** 4

**Summary:**

This paper proposes to study the so-called general cost OT problem, that is $\min_{\pi\in\Pi(P, Q)}F(\pi)$ where $F$ is a general functional and $|\Pi(P, Q)$ is the set of couplings between $P$ and $Q$. This is called general cost OT, because classical OT is a special case, taking $F(\pi) = \int c(x, y) d\pi$.
The authors derive a max-min reformulation for the approach as $\sup_{v}\inf_{\pi\in \Pi(P)} F(\Pi) - \int v d\pi(y) + \int v dQ(y)$, and then propose to parametrize the coupling $\pi$ as samples from $[x, T(x, z)]$ where $x\sim P$ , $z$ are samples from a Latent distribution like a Gaussian, and $T$ is a map that can itself be parameterized by a neural net. The potential $v$ is also parameterized by a NN.
In cases where the functional $F$ has a nice structure, the above max-min formulation can be estimated from random samples of $P, Q$ and $z$.

The authors then provide an error analysis for the method where $F$ is strongly convex.

The main application of the method discussed in the paper is to do optimal transport that is faithful to labels. Given two mixtures $P = \sum \alpha_n P_n$ and $Q = \sum \beta_n Q_n$, the authors want to estimate a transport plan between P and Q that should also map, as well as possible, each $P_n$ to $Q_n$. The corresponding cost function, a sum of energy distances between $T\\#P_n$ and $Q_n$, fits nicely into the proposed general cost framework, as it can be estimated from samples using a (costly) U-statistics, and is different from classical OT costs.

The experiments are on toy MNIST, KMNIST and fashion MNIST datasets, where the authors try to match each class. They compute the corresponding FID between mapped source and target dataset, and accuracy on the mapped set of a resnet trained on the target set.

**Strengths:**

The paper is very well written and easy to follow.
The algorithm developed in the paper is sound, and is an interesting method to estimate transport plans for general costs.
The theoretical results of the paper are interesting.
The problem of optimal transport with label faithfulness is also interesting for ML applications, and the proposed method is an elegant solution.

**Weaknesses:**

The main weakness of this paper is the experimental validation.
- the setup is very toyish: the datasets are toy datasets, and the labels are entirely unrelated: why match the digit '1' to 'trouser'? This is fine as a first toy experiment but there must be some more interesting ML applications where the labels from P and from Q have a relationship and are not paired randomly: having only this artificial experiment is underwhelming, and does not convince the reader that it is actually an interesting problem for machine learning.
- what does fig.3 show ? the description is far too short. Some methods are stochastic (i.e. T(x, z) with z random), how is the sample chosen?
- Same question for the metrics: how it is computed for stochastic outputs could be clearer. is it averaged over z?
- is FID computed per class or on the whole dataset? this should be clarified.
- all fonts are too small in the figures

Another point is that the cost function proposed in Prop.1 contains a quadruple sum over samples: a discussion about its variance would be welcome. Also, the proposition mentions that it is an estimator: in which sense?

**Questions:**

See above.


Misc. minor remarks:
- in the abstract, the use of transport "map" vs "plan" can be confusing
- Why write $\mathcal{X} = \mathcal{Y}$ in the notation? why bother with two spaces if they are the same.
- $\gamma$ is used for two different things on top of page 3
- "The performance of such methods in high dimensions is questionable": a reference to elaborate on this would be welcome.
- In eq.6, the second $\sup \inf$ can be removed to make it clearer what $\mathcal{L}$ is.
- middle of page 5: "it follows by solving (2)"  I think this refers to the unlabeled equation in corollary (2), not to eq.(2). Same with "Overall, Problem (2)" shortly after. It would probably be best to number the equation in corollary 7.
- "it does not always have a solution with each P_n exactly mapped to Q_n" a few more words about why this is the case would be nice.
- "we employ the Sinkhorn"
- what is the baseline accuracy of each resnet used in the experiments?

---

> ### Author Response · Authors · 2023-11-20
> **Thanks for your review**
>
> Thank you very much for your detailed analysis of our paper and thoughtful suggestions on paper improvements. We provide a response to your comments below.
>
> **Q1: the datasets are toy datasets, and the labels are entirely unrelated: why match the digit '1' to 'trouser'?  (...) there must be some more interesting ML applications where the labels from P and from Q have a relationship and are not paired randomly.**
>
> These experiments are for illustrative purposes only. Our goal here is to show that even when classes are *unrelated* ("1" to "trouser", etc.), our method still can learn an class-preserving map. For experiments on the *related* domains, when the labels from $\mathbb{P}$ and from $\mathbb{Q}$ have a relationship, please see **Appendix C.3**.
>
> In our paper, we also considered the application of our method to the well-known problem in biology called the *batch effect*, see **Appendix C.12**. This problem often occurs in genomic studies such as gene expression microarrays, RNA-seq, and proteomics analyses.
>
> For **new additional experiments** involving high-resolution images, please see the **general answer** provided to all reviewers and **Appendix E**.
>
> **Q2: what does fig.3 show ? the description is far too short. Some methods are stochastic (i.e. T(x, z) with z random), how is the sample chosen?.**
>
> Figure 3 provides the results of solving the dataset transfer problem with various methods. Each column shows the transfer result of a random (test) input $x\sim\mathbb{P}_{n}$ (first row) from a particular class ($n=0,1,\dots,9$). Each row show the results of transfer via a particular method in view. For methods which learn a stochastic map $T(x,z)$, we show their output $T(x,z)$ for a random noise $z$. *We added this description to Figure 3 in the revised paper.*
>
> **Q3: Same question for the metrics: how it is computed for stochastic outputs could be clearer. is it averaged over z?**
>
> Initially, we tried using multiple $z$ per $x$, generating $T(x,z)$, and averaging the metric over them. Later, we found that the result is the same if we sample only one $z$ per $x$. So we finally decided to sample a single $z$ per $x$ for simplicity. We have clarified this in the revised version. (Appendix C.1).
>
> **Q4: Is FID computed per class or on the whole dataset? this should be clarified? All fonts are too small in the figures**
>
> The FID was calculated on the entire dataset, not separately for each class. Following you suggestion, we clarified this in the **Metrics** paragraph of the paper (Section 5).
>
>
> **Q5: Another point is that the cost function proposed in Prop.1 contains a quadruple sum over samples: a discussion about its variance would be welcome. Also, the proposition mentions that it is an estimator: in which sense?**
>
> The reported quantity in Eq. (15) is the estimator of energy distance $\mathcal{E}$ in the sense that its expectation w.r.t. batch samples $X_n, Z_n, Y_n$ yields $\Delta \mathcal{E}^2(T_{\pi}\sharp (\mathbb{P}_n \times \mathbb{S}), \mathbb{Q}_n)$.
>
> It equals to $\mathcal{E}^2$ up to $T_\pi$-independent constant $C = - \frac{1}{2} \mathbb{E} \Vert Y_2 - Y_2' \Vert_2$, where $Y_2, Y_2'$ are independent copies from the target distribution $\mathbb{Q}_n$.
>
> Regarding the quadruple sum, we refer the reviewer to [1] for the analysis of similar quantities. Note that in practice we did not experience any problems with estimation of Eq. (15).
>
> [1] Sutherland et. al. Unbiased estimators for the variance of MMD estimators.

---

> ### Author Response · Authors · 2023-11-20
> **Part.2 (Minors)**
>
> **Q6.1. Why write $\mathcal{X}=\mathcal{Y}$ in the notation? why bother with two spaces if they are the same.**
>
> This is done for readability purposes and helps to better distinguish integrals over the input and target data spaces.
>
> **Q6.2 "The performance of such methods in high dimensions is questionable": a reference to elaborate on this would be welcome.**
>
> The general problem with these methods (wavelet/kernel-based, barycentric)  is actually the curse of dimensionality. Application of these methods bears similarity to application of kernel density estimators for approximating a probability distribution. It is not applicable to the distribution of images. This is a general problem with non-parametric setup. Regarding more theoretical justification, see [1]. They showed that as the dimensionality $d$ of data grows, the quality of the recovered map in the worst case behaves like $\approx N^{-c/d}$ for a constant $c$, where $N$ is the number of empirical samples.
>
> [1] Hutter et. al., Minimax rates of estimation for smooth optimal transport maps.
>
>
> **Q6.3: "it does not always have a solution with each $\mathbb{P}_n$ exactly mapped to $\mathbb{Q}_n$" a few more words about why this is the case would be nice.**
>
>
> The problem may occur with the existence of the solution since we try to map all class-specified distributions $\mathbb{P}_n$ to their counterparts $\mathbb{Q}_n$ with a **singe** stochastic map $T$. If, say, the source distributions are the same while the target ones are different, there is obviously no suitable map $T$.
>
> **Q6.4 what is the baseline accuracy of each resnet used in the experiments?**
>
> The accuracy of the ResNet18 classifiers is 99.17 for the MNIST and 95.56 for the USPS datasets. For the KMNIST and MNITST, the accuracy's are 99.39 and 97.19, respectively. We have included this information in Appendix C.1.
>
> **Concluding remarks**: Please respond to our post to let us know if the clarifications above suitably address your concerns about our work. We are happy to address any remaining points during the discussion phase; if the responses above are sufficient, we kindly ask that you consider raising your score.

---

### Author Response · Authors · 2023-11-20
**General Answer + paper revision**

Dear reviewers, we thank you for your provided feedback and suggestions on the paper improvements. Please consider the **updated paper and appendices**. The edits are highlighted by the *blue* color in the revised version of the submission. For a detailed overview of the provided updates, we kindly refer to our personal responses. The main update is summarized below.

**Q1: More example of cost functionals (nKa4, LT3k, iaG1, BH6f).**

To address questions regarding additional examples of general cost functionals [LT3k-Q3, BH6f-Q1] and experiments in high resolutions [nKa4-Q1, iaG1-Q5], we explored an additional problem.

We focused on the widely studied **paired** image-to-image translation problem. For this task, we introduced the following **pair-guided** cost functional $\mathcal{F}(\pi)\stackrel{def}{=}\int_{\mathcal{X}\times\mathcal{Y}}\ell(y,y^{\star}(x))d\pi(x,y),$ where we are provided with a paired dataset $(x_1,y^\star(x_1)),\dots, (x_N,y^{\star}(x_N))$ for training. Further details can be found in the newly added **Appendix E**. In that section, we present examples of the translation with our method at high resolutions (up to $512\times 512$) on Comic-faces-v1, Edges-to-Shoes and CelebAMask-HQ datasets.



We would be grateful if we could hear your feedback regarding our answers to the reviews and our revised submission. We are happy to address any remaining points during the remaining period.

---

### Meta-Review · Area_Chair_cmdy · 2023-12-06

**Metareview:**

The paper is well-written, displaying a commendable effort in articulating its content. It makes a notable algorithmic contribution to the estimation of transport plans for general costs, an area of increasing relevance in the field. The theoretical framework presented is both interesting and pertinent to current research trends. The primary weakness lies in the somewhat underwhelming numerical setups presented. Additionally, the algorithm, being an extension of the initial "Neural Optimal Transport" algorithm, suggests that the methodological contribution is somewhat limited. However, the authors have done a strong rebuttal, particularly in presenting new numerical data that enhance the paper's value. This effort in addressing the concerns raised and improving their work demonstrates a level of commitment and adaptability that is worthy of recognition. Hence, despite the mentioned shortcomings, the overall quality and contribution of the paper justify its acceptance. Despite several noted issues, I recommend the acceptance of this paper.

**Justification For Why Not Higher Score:**

I think the contributions are rather small.

**Justification For Why Not Lower Score:**

The new numerics added in the rebuttal is nice, and they did a nice job addressing issues.

---

### Decision · Program_Chairs · 2024-01-16

Accept (poster)